# Heterozygous loss of function of *IQSEC2*/*Iqsec2* leads to increased activated Arf6 and severe neurocognitive seizure phenotype in females

Matilda R Jackson[1,2], Karagh E Loring[1,2], Claire C Homan[2], Monica HN Thai[1], Laura Määttänen[3], Maria Arvio[3,4,5] , Irma Jarvela[6], Marie Shaw[2], Alison Gardner[2], Jozef Gecz[2,7] , Cheryl Shoubridge[1,2]

**Clinical presentations of mutations in the *IQSEC2* gene on the X-chromosome initially implicated to cause non-syndromic intellectual disability (ID) in males have expanded to include early onset seizures in males as well as in females. The molecular pathogenesis is not well understood, nor the mechanisms driving disease expression in heterozygous females. Using a CRISPR/Cas9–edited *Iqsec2* KO mouse model, we confirm the loss of *Iqsec2* mRNA expression and lack of Iqsec2 protein within the brain of both founder and progeny mice. Both male (52%) and female (46%) *Iqsec2* KO mice present with frequent and recurrent seizures. Focusing on *Iqsec2* KO heterozygous female mice, we demonstrate increased hyperactivity, altered anxiety and fear responses, decreased social interactions, delayed learning capacity and decreased memory retention/novel recognition, recapitulating psychiatric issues, autistic-like features, and cognitive deficits present in female patients with loss-of-function *IQSEC2* variants. Despite Iqsec2 normally acting to activate Arf6 substrate, we demonstrate that mice modelling the loss of Iqsec2 function present with increased levels of activated Arf6. We contend that loss of Iqsec2 function leads to altered regulation of activated Arf6-mediated responses to synaptic signalling and immature synaptic networks. We highlight the importance of IQSEC2 function for females by reporting a novel nonsense variant c.566C > A, p.(S189*) in an elderly female patient with profound intellectual disability, generalised seizures, and behavioural disturbances. Our human and mouse data reaffirm *IQSEC2* as another disease gene with an unexpected X-chromosome heterozygous female phenotype. Our Iqsec2 mouse model recapitulates the phenotypes observed in human patients despite the differences in the IQSEC2/Iqsec2 gene X-chromosome inactivation between the species.**

## Introduction

X-linked intellectual disability is a common, clinically complex disease arising from mutations in more than 140 genes on the X-chromosome (1), affecting between 1/600 and 1/1,000 males and a substantial number of females (2). X-linked inheritance is more complex than simply X-linked recessive or dominant (3) with both X-inactivation (including associated tissue specific selection) and the impact of individual mutations contributing to this complexity. In mammals, the sex determination system used is XX/XY, with dosage compensation in females as a result of random inactivation of one of the two X chromosomes in every cell. As a consequence, heterozygous females typically have a milder disease phenotype or are not affected. Despite this, there is a growing list of X-chromosome genes which are subject to X-inactivation or escape X-inactivation, including, for example, *PHF6, CLCN4, ALG13, ARX,* or *USP9X, DDX3X,* which display distinct phenotypes in males and females depending on the functional severity of the variant, as well as manifesting in a more severe female phenotype than the heterozygous state would predict (4, 5, 6, 7, 8, 9, 10). We contend that the IQ motif and Sec7 domain 2 protein (*IQSEC2*) (NM_001111125) (MIM 300522) is another X-chromosome disease gene in which we see a severe female phenotype because of heterozygous loss-of-function mutation.

We previously implicated *IQSEC2* as an X-linked intellectual disability (XLID) gene through identification of variants in affected males in four separate families (11). These missense variants were clustered around the Sec7 and IQ-like domains and resulted in reduced enzymatic activity (11). Clinical features within these non-syndromic XLID families included moderate to severe intellectual disability (ID) in all affected males, with variable seizures, autistic traits, and psychiatric problems (11). Since then, unbiased, high-throughput sequencing in ID and epilepsy cohorts have identified familial and increasingly de novo loss-of-function *IQSEC2* variants,

[1]Intellectual Disability Research, Adelaide Medical School, The University of Adelaide, Adelaide, Australia [2]Department of Paediatrics, Robinson Research Institute, University of Adelaide, Adelaide, Australia [3]Department of Child Neurology, Turku University Hospital, Turku, Finland [4]Joint Authority for Päijät-Häme Social and Health Care, Lahti, Finland [5]PEDEGO, Oulu University Hospital, Oulu, Finland [6]Department of Medical Genetics, University of Helsinki, Helsinki, Finland [7]South Australian Health and Medical Research Institute, Adelaide, Australia

Correspondence: Cheryl.shoubridge@adelaide.edu.au

typically leading to phenotypic outcomes, including severe ID with epileptic encephalopathy, and a high prevalence of speech development deficits and psychiatric features, including autistic spectrum disorder. Interestingly, these severe phenotypes are noted not only in affected males but also in affected, heterozygous females (12). The mechanisms contributing to the disease severity, particularly in heterozygous females is unknown and perplexing.

IQSEC2 is a guanine nucleotide exchange factor, which catalyzes exchange of GDP for GTP in a number of ARF superfamily of proteins. *IQSEC2* is highly expressed in the forebrain, specifically localized to excitatory synapses as part of the *N*-methyl-D-aspartate receptor (NMDAR) complex (13, 14). The exact role IQSEC2 plays at excitatory synapses remains unclear. Limited studies indicate a role in the activity-dependent removal of α-amino-3-hydroxy-5-methyl-4-isoxazolepropionic acid receptors (AMPAR) and activity-dependent synaptic plasticity (15, 16). Our own studies have shown that IQSEC2 also has a fundamental role in controlling neuronal morphology (17). However, there is currently no published research investigating the impact of loss or altered *Iqsec2* function on the development and resulting cognitive outcomes in any animal model. It is not certain if severe loss-of-function mutations in *IQSEC2* can be transmitted in the human setting, with only missense variants giving rise to milder non-syndromic features being maternally inherited. Hence, it was unclear if the loss of Iqsec2 function modelled in mice would survive into postnatal life, be reproductively viable or useful to model disease pathogenicity observed in humans. Here, we show that mice with the complete loss of function of *Iqsec2* by successfully targeting exon 3 using CRISPR/Cas9 technology survive into postnatal life and are viable. In this study, we investigate the effect of severe loss-of-function mutations driving the phenotype in patients, including the emerging female-specific phenotype using a mouse modelling the KO of *Iqsec2*.

We present an elderly female patient with profound ID and generalised seizures with a novel loss-of-function *IQSEC2* variant, providing important life span information for other patients diagnosed with this typically early onset neurodevelopmental disorder. We review the present literature of the growing number of females with loss-of-function variants in *IQSEC2*, who have a more severe phenotype than the heterozygous state would predict. In humans, the prevailing evidence suggests that *IQSEC2* escapes X-inactivation (18, 19); however, in mice, *Iqsec2* is subject to X-inactivation (20). Hence, the mouse modelling heterozygous KO of *Iqsec2* provides an opportunity to assess the impact of X-inactivation and altered *Iqsec2* gene dosage in females. Here, we show that the loss of Iqsec2 function in mice recapitulates key aspects of the human phenotype, irrespective of the X-inactivation status of the gene in the two species, highlighting that our understanding of the traditional X-chromosome inheritance with heterozygous female sparing needs to be revisited.

## Materials and Methods

### Animal generation

All animal procedures were approved by the Animal Ethics Committee of The University of Adelaide, Adelaide, Australia, and undertaken in accordance with their regulatory guidelines. Founding *Iqsec2* KO mice were generated by CRISPR/Cas9 by the South Australian Genome Editing facility (SAGE), University of Adelaide, Adelaide; details given below. Mice were maintained in the C57Bl/6N-Hsd background. Animals were of same sex and housed in individually ventilated cages, with sterile food and water available ad libitum. *Iqsec2* KO hemizygous male founder A was bred with a wild-type female to generate *Iqsec2* KO heterozygous progeny, which were subsequently bred with a wild-type stud male to generate *Iqsec2* KO hemizygous, *Iqsec2* KO heterozygous, and wild-type littermates. *Iqsec2* KO hemizygous males (n = 46) and *Iqsec2* KO heterozygous females (n = 153) were monitored and scored daily (from postnatal day [P] 14) for general health and welfare, appearance, weight, and the presence of seizure activity. In addition to standard food available ad libitum, crushed chow was soaked in sterile water and placed in an easily accessible feeding dish, which was refreshed daily.

### CRISPR/Cas9 guide design

Guides were designed by the SAGE facility, University of Adelaide, under the guidance of Professor Paul Q Thomas as part of a fee for service for the generation of CRISPR/Cas9 *Iqsec2* KO mice. The online tool (http://crispr.mit.edu/) was used to search for appropriate CRISPR guide sites targeting the removal of *Iqsec2* exon 3. The most appropriate guide was determined by total score, cut site in target gene, and number of off-target mismatches. Guides' sequences determined most suitable were upstream CRISPR guide 5′-TCTAGTGTACTCACTCAGTT-3′ and downstream CRISPR guide 5′-AGGCTGGAACTGGCGAAAAC-3′. The CRISPR/Cas9 complex will cause double strand breaks in intron 2–3 of *Iqsec2* and intron 3–4, causing exon 3 to be deleted by a process of non-homologous end joining. CRISPR gRNA generation, microinjections of zygotes, and transfer to pseudopregnant recipients were performed by the SAGE facility as previously described (21, 22, 23).

### Genotyping

A small segment of toe tissue was removed by sterile technique at P5 from all pups for genotyping and identification purposes. Genomic DNA was extracted as per the manufacturer's instructions for Phire Hot Start II DNA polymerase (Thermo Fisher Scientific). Genotyping PCR was performed using 10 μM forward and reverse primer pairs (Table S1), 2x Phire Tissue Direct PCR Master Mix, and made up to 20 μl with MilliQ water. Reactions were placed in a thermocycler for one cycle at 98°C for 5 min, 32 cycles at 98°C for 30 s, 62°C for 30 s, 72°C for 1 min, and one cycle at 72°C for 1 min. PCR products were held at 4°C before visualising on a 1.5% (wt/vol) agarose gel with 0.2 μg/ml ethidium bromide in TBE buffer (1.1 M Tris, 900 mM borate, and 25 mM EDTA, pH 8.3) alongside 1 kb+ molecular weight marker. Images were captured on a SynGene UV dock at 400 ms exposure on GeneSnap v7.05 for SynGene. DNA sequencing analysis was performed using SeqMan Pro version 10.1.2 (DNASTAR, Inc) against *Iqsec2* cDNA reference sequence NM_001114664.

### Analysis of Iqsec2 mRNA and Iqsec2 protein

Animals were humanely killed by cervical dislocation. Brain was dissected from the skull and cut into two halves sagittally along the

cerebral fissure. The right-hand side brain was separated and minced into cortex (n = 2, one each for protein and RNA) and cerebellum, snap-frozen in liquid nitrogen, and stored at –80°C pending analysis. RNA was extracted from 40 mg homogenised brain cortical tissue in TRIzol reagent and converted to cDNA using SuperScript RT (Thermo Fisher Scientific) as described previously (24). *Iqsec2* gene expression was performed using both RT-PCR and qPCR. RT-PCR was performed using 50 pmol forward and reverse primer pairs (Table S1), 20U Roche Taq DNA polymerase, FailSafe PCR 2X PreMix J (Epicentre), and made up to 50 µl with MilliQ water. Reactions were placed in a thermocycler for one cycle at 94°C for 2 min, 35 cycles at 94°C for 30 s, 57–60°C for 30 s, 72°C for 30 s, and one cycle at 72°C for 5 min. Images were captured as described above. qPCR was performed using TaqMan gene expression assay probes (Thermo Fisher Scientific) spanning exon 3–4 boundary (Mm02344188_m1), exon 11–12 boundary (Mm02344183_m1), and exon 13–14 boundary (Mm02344185_m1) with *gapdh* used as a housekeeper (Mm99999915_g1). Wild types were pooled and averaged based on sex (n = 4 female wild type, n = 5 male wild type). Each individual *Iqsec2* KO hemizygous male or heterozygous female sample was normalised to the averaged wild-type data for their respective sex, with resulting data displayed as relative *Iqsec2* expression to their respective sexed wild-type controls. Protein extraction, SDS–PAGE, and Western blot analysis of protein levels were performed as described previously (17). The primary antibodies were rabbit anti-IQSEC2 (1:2,000) as previously described (17), rabbit anti-IQSEC1 and rabbit anti-IQSEC3, both used at 1:1,000 (Invitrogen), and mouse $\beta$-actin (AC-74; 1:20,000; Sigma-Aldrich A2228). Secondary antibodies from DAKO (Santa Clara) were goat antimouse HRP (1:2,000 P0447) and goat antirabbit HRP (1:2,000 P0448). Images were imported into Image Studio (Li-Cor Biosciences), and band intensities of Iqsec proteins were normalised to their respective $\beta$-actin loading control, and where required were harmonized across multiple immunoblots using a consistent control sample. Each individual *Iqsec2* KO hemizygous or heterozygous sample was normalised to the averaged pooled wild-type data for their respective sex, with relative intensities presented (n for each as described in figure legends).

## Behavioural assessment

*Iqsec2* KO heterozygous and wild-type females underwent monthly behavioural testing from one to 6 mo of age (n = 4 *Iqsec2* KO heterozygous and n = 3 wild-type controls at 1 mo, n = 8 *Iqsec2* KO heterozygous, and n = 6 wild-type controls at all other time points) as previously described (25).

## Neuroanatomy

The left-hand side brain, separated along the cerebral fissure, was fixed at 4 degrees in 10% neutral buffered formalin overnight, before being washed three times in cold PBS, and stored in 70% ethanol at 4 degrees. The samples were processed and paraffin-embedded by the Adelaide University Histology Department. Semi-serial sections, measuring 10 µm thick, were collected using a strategy of mounting every fifth serial section across five slides (series 1–5) with up to five replicates of this strategy per sample

(A–E) to span the breadth of the mouse brain in sagittal or coronal orientation. The sections were stained by haematoxylin and eosin or Nissl and scanned using a Hamamatsu NanoZoomer 2.0-HT whole slide imager (Meyer Instruments). Images were imported into ImageJ (Fiji; version 2.0.0-rc-59/1.51k, build fab6e1a004) for processing and measurement.

## Multielectrode array

Cortical neuronal were isolated from embryonic day (E) 17.5 *Iqsec2* KO heterozygous female (n = 12) and wild-type littermates (n = 7 female) as per Hinze *et al* (17). Neuronal suspensions were plated at 2.97 × 10$^5$ cells/well on 0.1% polyethyleneimine/20 µg/ml laminin-coated 24-well glass bottomed multielectorde array (MEA) plate (product: 24W300/30G-288; MultiChannel Systems). After 21 d in culture, 15-min recordings were captured using MultiScreen (version 1.5.9.0; MultiChannel Systems) at a sampling rate of 20,000 Hz and 1,000 ms baseline duration. Spike binning was performed at 100-ms intervals, with minimum burst duration set at 50 ms, with a minimum spike count in burst set at four spikes. Captured data were uploaded and exported using MultiAnalyser (version 1.2.90; MultiChannel Systems). Data obtained from each individual *Iqsec2* heterozygous female embryo was normalised to the averaged wild-type data.

## G-LISA Arf6 activation assay

Protein was extracted from snap-frozen cortical tissue of wild-type male mice (n = 4), KO male mice (n = 6), wild-type female mice (n = 5) and HetKO female mice (n = 9) following the manufacturer's instructions for use in a G-LISA Arf6 Activation assay Biochem Kit (absorbance based) (Cytoskeleton). The levels of activated Arf6 measured in the cortical tissue were from mice ranging in age from 2 to 9 mo. Each sample was measured with an n = 4 replicates. Within each assay, the levels of activated Arf6 measured in the wild-type animals for each sex was set to 1, and values for each of the KO or HetKO samples were determined relative to these age and sex-matched wild-type controls. The relative levels of activated Arf6 for all WT animals measured in a single assay were used to normalise activated Arf6 levels across multiple assays. An aliquot of protein from each cortical sample analysed in the GLISA assays was also prepared for SDS–PAGE and Western blot analysis and probed for Iqsec2 as described (17) and Arf6 protein abundance using polyclonal Arf6 antibody (PA1-093; Thermo Fisher Scientific) and quantitated as indicated above for Iqsec2 protein abundance.

## Molecular analysis of IQSEC2 variant

The screening protocols were approved by the Women's and Children's Health Network Human Research Ethics Committee and the Human Ethics Committee of The University of Adelaide, Adelaide, Australia (approval number REC2361/03/2020) and conforms with the principles set out in the WMA Declaration of Helsinki and Australian National Statement on Ethical Conduct in Human Research (2018). Informed consent was obtained from carers of the patient, including consent to publish images. DNA from the affected female was whole-exome sequenced on an Illumina HiSeq2500 by the Australian Genome Research Facility. Reads were mapped to

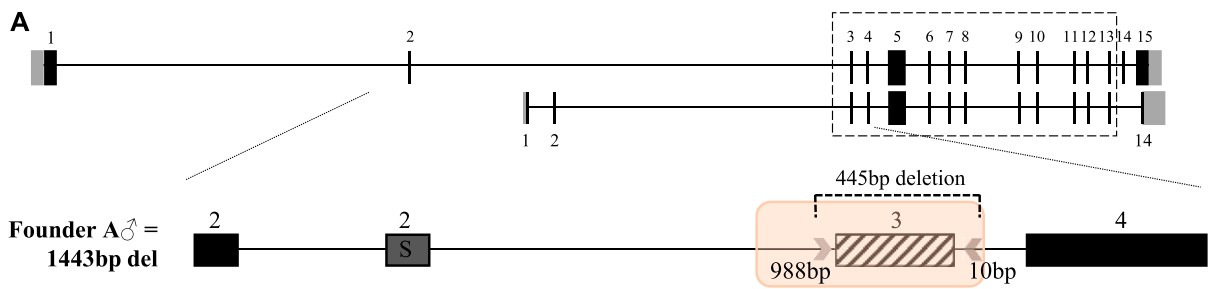

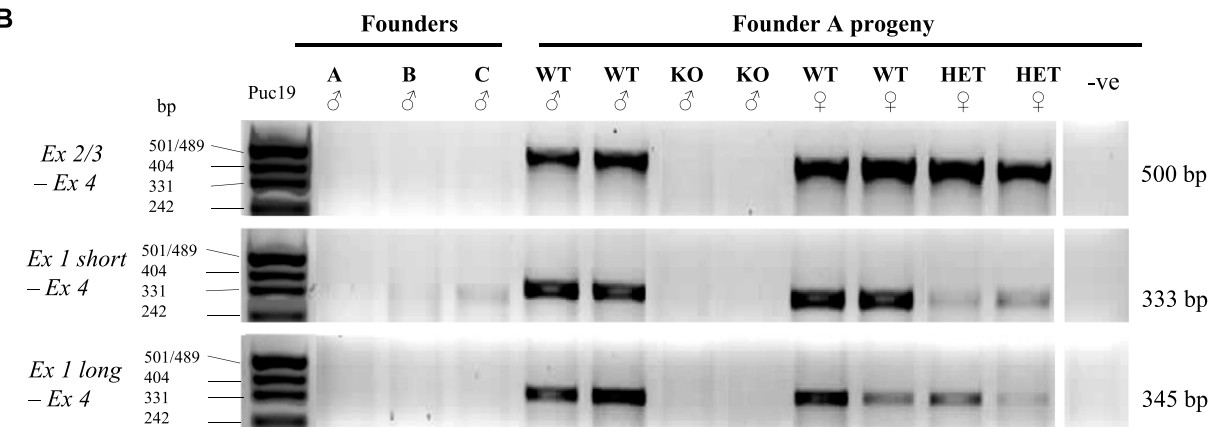

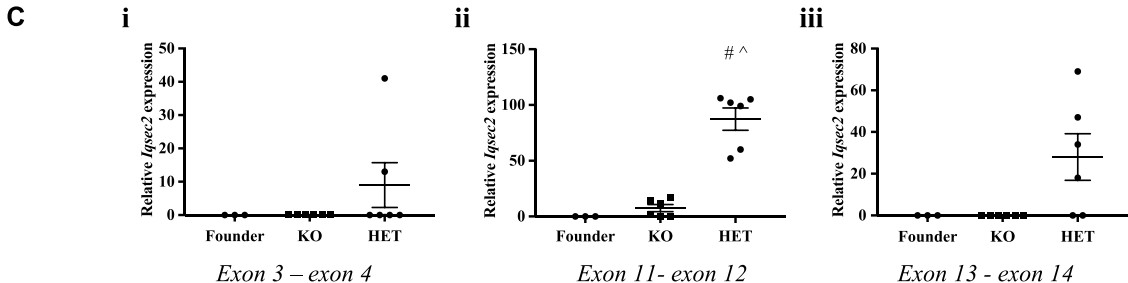

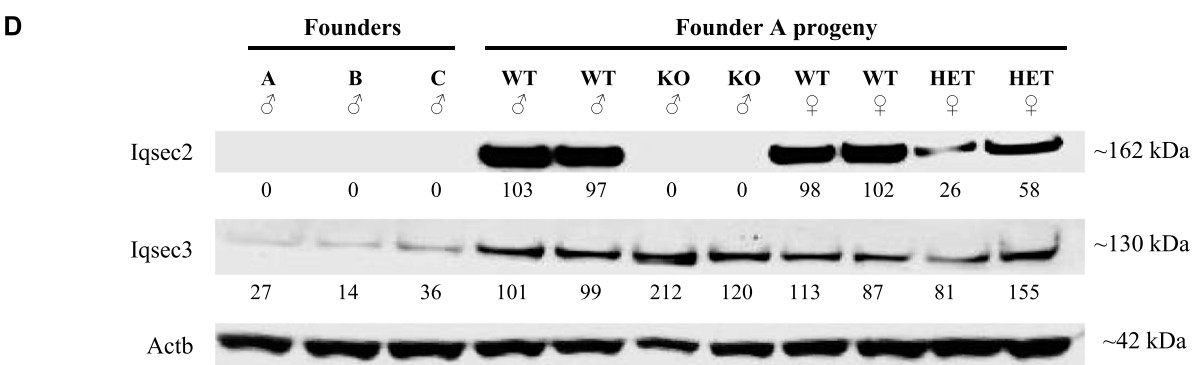

**Figure 1. CRISPR/Cas9 targeting of *Iqsec2* resulted in absent (KO males) or reduced (KO HET females) *Iqsec2*/Iqsec2 expression.**
**(A)** Schematic of the exon–intron structure of *Iqsec2* long (NM_001114664) and short (NM_001005475) isoforms, with the dashed box indicating consensus sequence. Zoomed-in schematic of *Iqsec2* exon 2–4 (exon 2 long = 2L and exon 2 short = 2S) with CRISPR guides (arrows) flanking exon 3 (diagonal line fill), resulting in a predicted 445-bp deletion. Actual deletion size (highlighted by an orange box) shown flanking CRISPR guide putative cut sites in Founder A, with sex and total deletion size shown on left-hand side. **(B)** RT-PCR amplification of exon 2/3 boundary, exon 1 (short isoform), and exon 1 (long isoform) to exon 4 of 3 male founders, and subsequent progeny from founder A. **(C)** qPCR of three founder males (grouped) and subsequent progeny from founder A. Results are expressed as mean relative expression (±SEM; n = 3

the human genome (hg19) using BWA-MEM (26) and mapping refined using Genome Analysis Toolkit version 3.5 (27). Mapping achieved a minimum median target coverage depth of 49 reads/sample and covered 87.67% of intended targets with at least 20 reads. Single-nucleotide variants and small insertions and deletions were called by the genome analysis toolkit haplotype caller version 3.5 (27). Whole-exome sequencing data are available upon request.

All variants were annotated for allele frequency, clinical significance, locus identity, and likely pathogenicity using ANNOVAR (28).

### Statistical analysis

The statistical significance ($P < 0.05$) of the difference between means of each strain, namely, *Iqsec2* KO hemizygous males, *Iqsec2* KO heterozygous females, and their respective age-matched control littermates, was determined using multiple statistical means. A one-way ANOVA followed by Tukey's HSD post hoc test was used for qPCR and MEA analysis, whereas a two-way ANOVA followed by Tukey's HSD was used to assess behavioural differences across time. A two-tailed, unpaired *t* test was used to compute statistical significance of the difference between means for the GLISA analysis, and when wild-type littermates were omitted from statistical analysis (seizure propensity). All data analyses were performed using GraphPad Prism version 7 (GraphPad Software Inc.).

## Results

### Generation of loss-of-function *Iqsec2* KO mice by CRISPR/Cas9 deletion of exon 3

Generation of a knockout (KO) mouse line to model the loss of Iqsec2 function was achieved by targeting exon 3 of *Iqsec2* for deletion by CRISPR/Cas9 editing (17). The sequence and targeting of CRISPR guides to remove exon 3 in both isoforms are detailed in Fig S1. Exon 3 is invariable between the two main isoforms of *Iqsec2*, with exons 3–13 comprising consensus sequence (Fig 1A). Injection of the CRISPR/Cas9 guides was performed as a fee for service (South Australian Genome Editing facility, University of Adelaide, Adelaide) (21). PCR amplification of genomic DNA (Fig S2A) and subsequent breakpoint mapping and sequencing of amplicons demonstrate that Founder A (male) had exon 3 removed with neighbouring exon 2 (short and long isoforms), exon 4, and exon 5 unaffected by the deletion, demonstrating successful targeted deletion of exon 3 (Fig 1A). Although exon 3 did not amplify in any of the four founders generated (Fig S2B), founders B and C (males) had larger deletions than expected (Fig S2A), both impacting exon 4, and

founder D (female) had a homozygous deletion of exon 3 (Fig S2C). The homozygous loss of *IQSEC2* has not been reported in the human population. Given the increasing incidence of mutations in girls (heterozygous) with early-onset seizure phenotypes, it was not unexpected that this female homozygous KO mouse was found dead early in postnatal life, negating the opportunity to collect samples for expression analysis, or attempt breeding. This animal was not included in any further analysis. Overall, the editing of all founders extended past the recognised CRISPR/Cas9 guide cut sites in both directions by nonstandard amounts, with larger deletion sizes of this X-chromosome region noted in males. This finding demonstrates that CRISPR/Cas9, although an effective genome editing tool, requires careful validation.

To confirm that deletion of exon 3 by CRIPSR/Cas9 editing resulted in loss of *Iqsec2*/Iqsec2, we analysed the gene expression and protein level from cortical brain tissue in which *Iqsec2* is highly expressed during postnatal life. We demonstrate in the three founder males and progeny of founder A that *Iqsec2* expression was reduced or absent when detected by RT-PCR (Fig 1B) and significantly reduced compared with sex-matched wild-type progeny when analysed by qPCR (Fig 1C). Founder A and founder B had no detectable *Iqsec2* expression by either analysis. The negligible expression levels of *Iqsec2* short isoform detected for founder C in the RT-PCR analysis was not replicated by qPCR. *Iqsec2* KO hemizygous male progeny from founder A also had no detectable *Iqsec2* expression by RT-PCR (Fig 1B) but demonstrated minimal *Iqsec2* expression by qPCR with the probe spanning across the exon 11–12 boundary Fig 1C. (ii) This result is consistent with very low levels of transcript (~5% of normal) being present before nonsense-mediated mRNA decay. *Iqsec2* KO heterozygous female progeny showed that *Iqsec2* expression by RT-PCR for both the short and long isoforms were reduced to less than half the levels of wild-type female controls (Fig 1B). Similarly, reduced *Iqsec2* expression in these females was also noted by qPCR, with expression levels dependent on the probe used (range 0–79%, mean 35%), but were still significantly elevated above both founder males and *Iqsec2* KO hemizygous male progeny (Fig 1C). We note that expression levels were quite varied between individual heterozygous females and cannot discount differences due to levels of X-inactivation in these animals. The three male founders had no discernible Iqsec2 protein in the cortex (Fig 1D). Similarly, there was no discernible Iqsec2 protein in KO hemizygous male progeny and reduced levels of Iqsec2 protein (~half of the wild-type control levels) in KO heterozygous female progeny (Fig 1D). Off-target analysis of CRISPR/Cas9 editing using computational tools (CRISPR design tool by MIT: http://crispr.mit.edu/ and COSMID: https://crispr.bme.gatech.edu) identified that the majority (92%) of predicted off-targets were located in regions that did not harbour genes or impacted intronic regions within genes and were unlikely to effect the coding region

founders, n = 6 *Iqsec2* KO hemizygous males (KO), n = 6 *Iqsec2* KO heterozygous females [HET]) normalised to wild types, which were pooled and averaged dependent on sex (n = 4 female wild type, n = 5 male wild type). **(C)** TaqMan gene expression assay probes spanning (i) exon 3–4 boundary, (ii) exon 11–12 boundary, and (iii) exon 13–14 boundary with *gapdh* used as a housekeeper. **(D)** Western blot analysis of Iqsec2 and Iqsec3 expression in three founder males and subsequent progeny from founder A. Blots were imported into Image Studio (Li-Cor Biosciences) and band intensities normalised to their respective beta-actin (Actb) loading control. Wild types were pooled and averaged dependent on sex (n = 2 female wild type, n = 2 male wild type). White spaces indicate a cropped image. # indicates significant difference between HET and founder, $P < 0.0001$ one-way ANOVA, Tukey's HSD, ^ indicates significant difference between HET and KO, $P < 0.0001$; one-way ANOVA, Tukey's HSD.

of the genome (Table S2). For the 31 genes identified to potentially be impacted by off-target effects, 28 were reported with high numbers of mismatch (n = 4). In contrast, *Iqsec3* was the only gene predicted to be impacted by either CRISPR guide that was identified by both computational tools (Table S2). *IQSEC3* is highly expressed within multiple regions of the brain, including the cortex. Our analysis of the three founder males demonstrates that Iqsec3 protein levels in the cortex were reduced (27%, 14%, and 36% of wild-type, respectively) but were normalised to wild-type levels in subsequent founder A progeny (Fig 1D). Taken together, these data suggest successful outbreeding of the potential off-target effects.

To investigate if the loss (or partial loss) of Iqsec2 protein in the *Iqsec2* KO mice elicited a compensatory effect by other members of the Iqsec protein family, we measured the protein abundance of Iqsec1 and Iqsec3 in cortical samples of the wild-type and *Iqsec2* KO mice by immunoblot. The levels of Iqsec1 protein were very low compared with the ready detection of Iqsec3 protein in the same samples and were not robust enough for semiquantitative analysis. Despite this, we did not see any empirical evidence of a consistent or stronger signal in the *Iqsec2* KO animals. In the case of Iqsec3, there was no significant increase in protein abundance in the

brains of *Iqsec2* KO hemizygous male or KO heterozygous female mice compared with wild-type sex-matched mice (Fig S3). Hence, we demonstrate that Iqsec protein family members are unlikely to provide a compensatory role to ameliorate the loss or partial loss of Iqsec2.

## *Iqsec2* KO male and female mice present with spontaneous seizures

We observed severe spontaneous seizures in both *Iqsec2* KO hemizygous male and heterozygous female mice modelling loss of Iqsec2 function. We saw a combination of four seizure subtypes that although distinctive in appearance, were often observed in a single seizure episode in both sexes. The seizures included (i) sudden onset of irregular generalised clonic jerks, where the mouse demonstrated involuntary, uncontrolled, unilateral head movements (Video 1); (ii) repetitive forelimb clonus that commenced with intermittent clonic jerking of the head and forelimbs, which then became rhythmic, associated with tonic posturing of the forelimbs, evolving to rearing and generalised tonic–clonic activity lasting ~60 s (Video 2); (iii) uncontrolled convulsions with bilateral forelimb

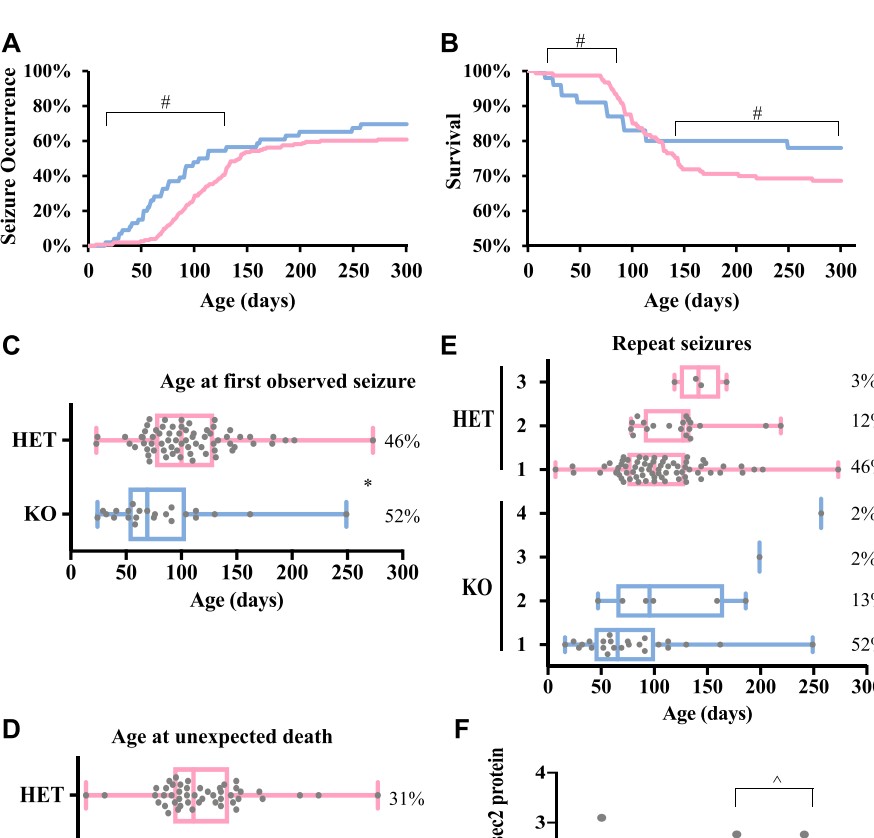

**Figure 2.** ***Iqsec2* KO hemizygous males and heterozygous females exhibit spontaneous seizures and reduced survival, which was not observed in their wild-type control littermates.**
The total number of animals phenotyped include KO; n = 46 (blue) and HET; n = 153 (pink). **(A, B, C, D, E)** Percentage seizure occurrence and (B) survival presented at daily intervals from birth, with both further subclassified as (C) age at observed first seizure, (D) age at unexpected death, and (E) occurrence of repeat seizures presented as median (±min/max), where each dot represents an individual animal. Unexpected death was classified as humane euthanasia or found dead presumed because of seizure or status epilepticus and does not include those individuals taken for experimental end point. These data do not include any movement phenotypes observed. **(F)** *Iqsec2* heterozygous females (Het/pink; n = 13) have reduced levels of Iqsec2 protein compared with female wild-type animals (WT/grey; n = 11). Mean (±SEM) data presented. The animals with observed seizures are denoted as stars. There were no significant differences in Iqsec2 protein abundance between male (WT/Black; n = 10) and female wild-type animals (WT/grey; n = 11). * indicates significant difference between KO males and HET females, *P* < 0.05, two-tailed, unpaired *t* test, # indicates *P* < 0.0001, two-tailed, paired *t* test, ^ indicates *P* < 0.05 between HET/KO and female WT controls, two-tailed, unpaired *t* test.

outward stretching were also noted which commenced with the sudden onset of irregular generalised clonic jerks followed by hypermotor activity, which lasted ~5 s before ceasing (and often reinitiating; Video 3); and (iv) and full body tonic–clonic seizures that started with symmetrical tonic extension of both forelimbs and hindlimbs, which develop into rhythmic generalised clonic activity after ~15 s, continuing for an additional 30 s before ceasing (Video 4). Seizure episodes were frequently accompanied by twitching ears, a straight tail, and an increase in facial grooming/washing pre- and post-seizure occurrence. *Iqsec2* KO mice that were found dead in their cage were classified as having died because of either a seizure or status epilepticus, as no wild-type control littermates were found dead in this study.

In concordance with the clinical variability noted in both male and female human patients with loss-of-function mutations, we measured large variations in age of onset, seizure severity, and progression amongst individual *Iqsec2* KO mice. The proportion of *Iqsec2* KO hemizygous males exhibiting seizures (all sub-types combined) from birth to 4 mo of age (7–54%, respectively) was

significantly increased when compared with heterozygous females (2–37%, respectively; Fig 2A). However, after 5 mo of age, the proportion of *Iqsec2* KO mice exhibiting seizures plateaued, with males ranging from 57 up to 65% at the study end point (~300 d of postnatal life) and females ranging from 54 to 60% across the same time period. The survival from birth to 3 mo of age in *Iqsec2* KO hemizygous males (87%) was significantly decreased compared with heterozygous females (91%; Fig 2B). From 4 mo of age, *Iqsec2* KO males reached a plateau (80%), whereas survival of heterozygous females continued to significantly decline until 8 mo of age (69%). The first observed seizure occurred with similar timing in both *Iqsec2* KO male and female mice, at postnatal day (P) 29 and P23, respectively (Fig 2C), with the first unexplained death occurring at P16 and P7, respectively (Fig 2D). The majority of male (52%) and female (46%) *Iqsec2* KO mice were observed to have only one seizure. The proportion of mice observed to have two or more seizures spanning their postnatal life were similar between male and female KO mice (13% and 2% versus 12% and 3%, respectively; Fig 2E). The levels of Iqsec2 protein abundance in cortical tissue

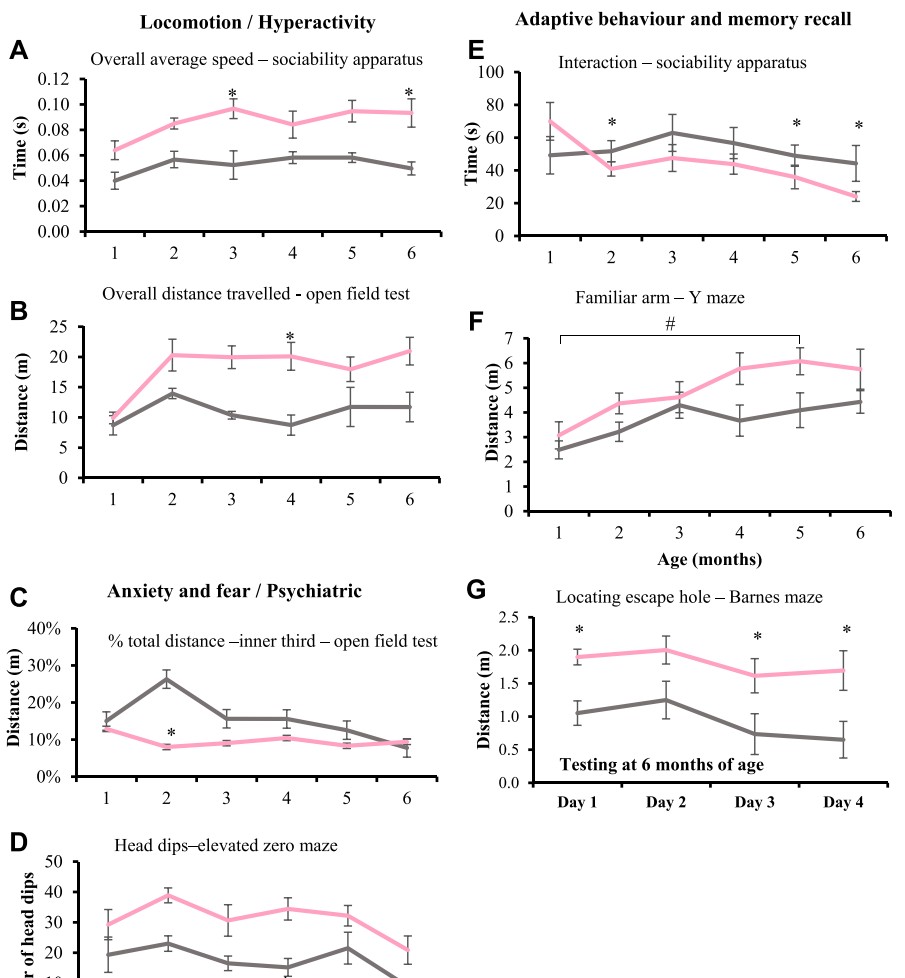

**Figure 3.** ***Iqsec2* KO heterozygous females display altered anxiety, increased locomotor activity and reduced spatial learning and memory.**

**(A, B, C, D, E, F, G)** Behavioural tests undertaken at monthly intervals between 1 and 6 mo of age show that *Iqsec2* KO heterozygous females (HET/pink) (n = 4 at 1 mo; n = 8 at 2–3 mo, n = 7 at 4–6 mo) compared with their wild-type female controls (WT/grey) (n = 3 at 1 mo; n = 6 at 2–6 mo) demonstrate (A) increased speed across multiple apparatus (sociability apparatus shown), (B) increased exploratory behaviour in the open field test, (C) increased anxiety in open field test, (D) decreased fear response in the elevated zero maze, (E) reduced total interaction time in the sociability apparatus regardless of familiar or novel cage occupant, (F) decreased novel recognition in the Y-maze, (G) and reduced spatial learning in the Barnes maze (conducted at 6 mo of age). Mean (±SEM) data presented, where * indicates significance between HET/KO and WT controls, # indicates significant between HET time points, two-way ANOVA with Tukey's HSD.

were reduced in heterozygous female mice compared with wild-type female littermates (1.8 fold, *P* = 0.0264) as measured by semiquantitative immunoblot (Fig 2F). Iqsec2 protein abundance was not impacted by the presence of an observed seizure (indicated by stars instead of circles).

## *Iqsec2* KO heterozygous female mice demonstrate altered behavioural phenotyping

Given the striking similarity in the seizure phenotype displayed by the *Iqsec2* KO hemizygous male and heterozygous female mice, coupled with the marked overlap in phenotypic outcomes in loss-of-function male and females patients in the human setting, we contend that the *Iqsec2* KO heterozygous females provide a representative model for this disorder. Interestingly, the male KO progeny were able to breed and transmit the loss-of-function *Iqsec2* mutation with minimal difficulty. In contrast, the *Iqsec2* KO heterozygous females displayed reduced breeding success (Fig S4). However limited, this indicates that a loss-of-function mutation

in *Iqsec2* is able to be transmitted, at least in mice. Due largely to the limited breeding success of our *Iqsec2* KO heterozygous females, generating the required number of age appropriate KO males for behavioural testing was challenging and precluded testing in hemizygous males. Hence, we undertook a battery of behavioural tests at monthly intervals up to 6 mo of age in the *Iqsec2* KO heterozygous females compared with female wild-type controls.

*Iqsec2* KO heterozygous females demonstrate an increased locomotor activity and exploratory behaviour. *Iqsec2* KO heterozygous females exhibit reduced neuromuscular strength at 3, 5, and 6 mo of age using the inverted grid test (Fig S5A). Although the difference in overall performance by each genotype on the apparatus was significant (*P* = 0.0031), the variability amongst individuals meant that significance was not reached at any individual time point. Hyperactivity was indicated by a significant (*P* < 0.0001) increase in the overall average speed (Fig 3A; sociability apparatus shown), and overall total distance travelled (Fig 3B; open field test shown) compared with wild-type littermates on multiple apparatus.

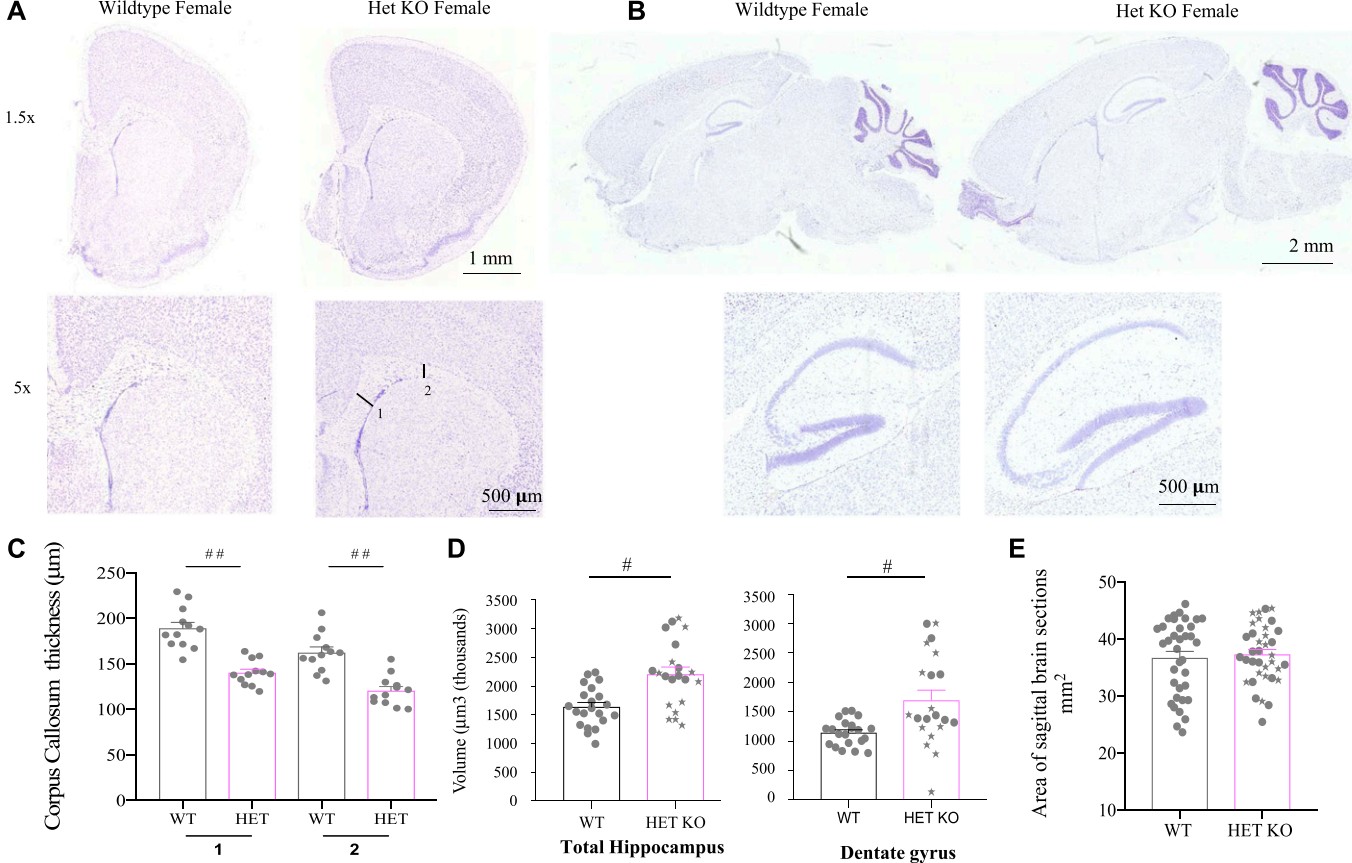

**Figure 4. Neuroanatomy changes in *Iqsec2* KO mouse brains.**
**(A, B)** Nissl staining of brain sections in both (A) coronal and (B) sagittal orientations from adult wild-type and heterozygous KO females demonstrate there is no gross disturbance to brain morphology. Scale bars shown for each set of pictomicrographs. **(C)** The thickness of the corpus callosum (CC) measured where the (1) start and (2) end of the cingulum intercepts the CC in three coronal sections for each of n = 4 animals per genotype is significantly thinner in heterozygous female (HET/pink) mice compared with wild-type female (WT/Grey) mice. **(D)** Heterozygous females have increased total hippocampal and dentate gyrus volume compared with wild-type littermates (n = 4 each). **(E)** The area of the brain was measured in animals from 60 to 155 d postnatal age in sagittal sections in HET/pink and WT/grey females (a total 36 sections measured per genotype: nine sections each for n = 4 animals). The *Iqsec2* HetKO animals with observed seizures are denoted by stars. Mean (±SEM) data presented where # indicates *P* < 0.001, and # # indicates *P* < 0.0001, two-tailed, unpaired *t* test between wild-type and heterozygous females.

Altered anxiety and fear response was also noted in *Iqsec2* KO female mice, with a significant increase in percentage total distance travelled in the inner third of the open-field apparatus (Fig 3C and *P* = 0.0006), increased number of head dips on the elevated zero maze (Fig 3D and *P* < 0.0001), and an initial increase in percentage total distance travelled in the open arms of the elevated zero maze (Fig S5B and *P* = 0.0007). This is in comparison with wild-type females, which from 4 mo of age travelled a greater percentage of total distance on the open arms of the elevated zero maze, suggesting habituation (Fig S5B).

Normal social interaction and memory recall is demonstrated by mice choosing to interact with a mouse rather than an empty cage and a preference to interact with a novel mouse over a familiar mouse when tested in a three-chamber sociability apparatus. The *Iqsec2* KO heterozygous females displayed autistic-like behaviour evident by a significant decrease in the total overall time of interaction, regardless of the other cage occupant (Fig 3E and *P* < 0.03). In addition, the behaviour of the *Iqsec2* KO heterozygous females suggests reduced learning capacity and spatial memory retention in contrast to wild-type females. *Iqsec2* KO heterozygous

females had a significant preference toward exploring the familiar arm of the Y-maze increasing over time (Fig 3F and *P* = 0.0261) and took significantly longer and travelled further distance to find the escape hole in the Barnes maze (Fig 3G and *P* < 0.0265).

## Altered neuroanatomy in *Iqsec2* KO heterozygous female mice

Nissl stain of both the coronal (Fig 4A) and sagittal (Fig 4B) orientations demonstrate there is no gross disturbance to the overall anatomical structure of the brain in heterozygous female compared with wild-type mice. Emerging evidence suggests that severely affected patients with *IQSEC2* loss-of-function mutations display a phenotype that includes thinning of the corpus callosum. We examined the thickness of the corpus callosum at two distinct points, each in three semi-serial coronal sections from an n = 4 animals per genotype (sections relative to the reference Allen Brain Atlas, Mouse, P56, and coronal images 44–46 of 132). Using this approach, we demonstrate that the thickness of these regions was significantly diminished in the heterozygous female mice compared with the wild-type female mice (Fig 4C). In line with the increase in

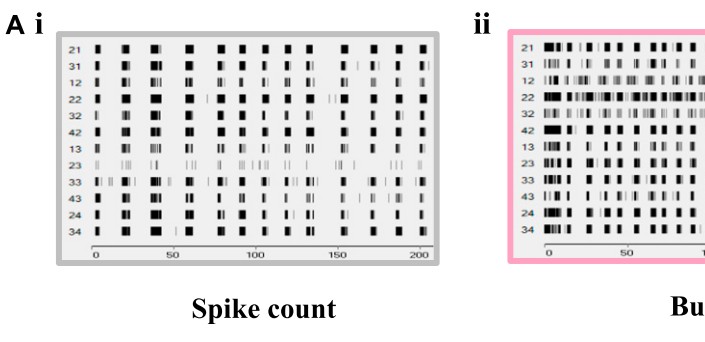

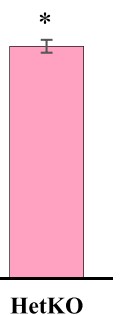

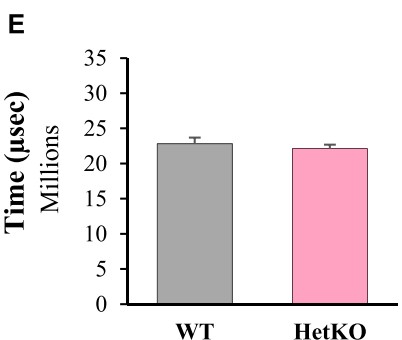

**Figure 5.  Embryonic (E)17.5 cultured cortical neurons from *Iqsec2* heterozygous females (HET/pink; n = 449 electrodes from n = 12 embryos) exhibit hallmarks of immature synaptic networks when compared with their respective wild-type (WT) control littermates (WT female/grey n = 256 electrodes from n = 7 embryos).**
**(A, B, C, D)** Representative raster plots for (A.i) WT female and (A.ii) HET female after 21 d in culture. Quantitatively, HET cultures showed an increased (B) spike count, (C) burst count, and (D) mean burst duration compared with wild-type control littermates. Mean (±SEM) data presented, where * indicates significance between HET and WT control, two-tailed, unpaired *t* test , where *P* < 0.05.

hyperactivity and anxiety noted in our *Iqsec2* KO heterozygous females, we observed an increase in total hippocampal volume in heterozygous KO female mice (34%, *P* = 0.0008 HET versus WT females) and an increase in the volume of the dentate gyrus (48%, *P* = 0.0006 for HET females) shown in Fig 4D (sections relative to the reference Allen Brain Atlas, Mouse, P56, sagittal images 9–10 of 21). Despite these discreet changes to specific neuroanatomical regions of the brain, our analysis indicates there were no significant differences in the area of the brain sections measured between wild-type females and the heterozygous KO females (Fig 4E) (sections relative to the reference Allen Brain Atlas, Mouse, P56, sagittal images 11–12 of 21).

### *Iqsec2* female mice display hallmarks of immature synaptic networks ex vivo

To investigate the impact of Iqsec2 on the activity of synaptic networks, we analysed cultured cortical neurons using a multi-electrode array. Neurons isolated from individual *Iqsec2* KO heterozygous female embryos grown for 21 d in culture display hallmarks of immature synaptic networks when compared with neurons from wild-type littermates (Fig 5). Burst activity and bursting behaviour is considered one of the most important properties for analysing synaptic plasticity and information processing within the central nervous system. Wild-type cultures demonstrated consistent, evenly spaced bursts of closely clustered action potentials, suggestive of a highly synchronised culture (Fig 5A.i). In contrast, *Iqsec2* KO heterozygous cultures showed aberrant synchronicity; with action potentials not consistently clustered in large bursts, but included smaller, randomly spaced events (Fig 5A.ii). In line with this observation, cultures from heterozygous females exhibited significantly elevated spike count (121%; Fig 5B and *P* < 0.0001), burst count (113%; Fig 5C and *P* = 0.0002), and mean burst duration (114%; Fig 5D and *P* = 0.003), but no difference in mean burst interval (Fig 5E) when compared with sex-matched wild-type controls. It must be noted that large variations existed in *Iqsec2* KO heterozygous cultures, with some embryos demonstrating burst patterning similar to that of their wild-type counterparts.

### Loss of *Iqsec2* function leads to an increased level of activated Arf6 in cortical tissues

There are multiple ArfGEFs, each with a conserved Sec7 domain responsible for catalysing nucleotide exchange and activating members of the small G protein Arfs. Hence, it was not clear what impact the loss-of-function of a single ArfGEF would have on regulating activated Arf-mediated responses to synaptic signalling in vivo. As Iqsec2 is an ArfGEF particularly for the small GTPase Arf6, we analysed the levels of activated (or GTP bound) Arf6 in cortical tissues from individual mice of each genotype. There was no change in the level of Arf6 activation with increasing postnatal age (range 2–9 mo), and although the levels of activated Arf6 were higher in wild-type females relative to wild-type males, this difference was not significant (1.5 fold; Fig 6A). In contrast, cortical tissues from heterozygous females exhibited significantly elevated levels of activated Arf6 (2.6 fold; *P* = 0.0186) when compared with sex-

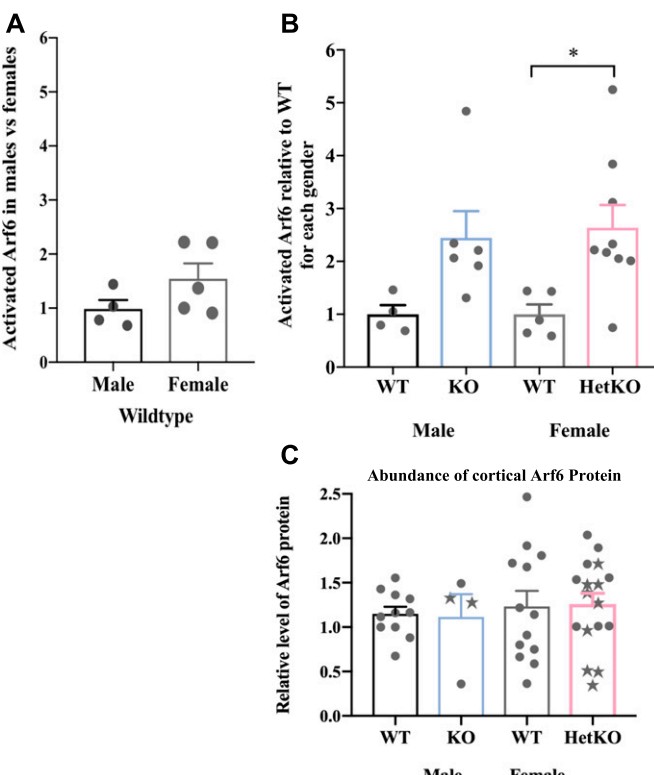

**Levels of Activated Arf6 in cortical tissue**

**Figure 6. The levels of activated Arf6 in cortical tissue are elevated because of *Iqsec2* KO.**
**(A)** Biochemical assays to measure the levels of activated Arf6 (G-LISA) undertaken in cortical tissues of animals across postnatal development between 2 and 9 mo of age show that (A) the levels of activated Arf6 in wild-type male mice (WT/black) (n = 4) are elevated in age-matched wild-type female mice (WT/grey) (n = 5). **(B)** *Iqsec2* KO hemizygous males (KO/blue, n = 6) and heterozygous KO females (Het/pink; n = 9) both display increased levels of activated Arf6 compared with the sex-matched wild-type controls listed above. **(C)** The abundance of Arf6 protein measured by immunoblot was not significantly different between any genotype groups. The *Iqsec2* KO animals with observed seizures are denoted by stars. Mean (±SEM) data presented, where * indicates significance between WT female control and HET/KO, *P* < 0.05, 2-tailed, unpaired *t* test.

matched and age-matched wild-type controls (Fig 6B). A similar outcome was observed in cortical tissues from hemizygous males with an elevated, although not statistically significant, level of activated Arf6 (2.45 fold; *P* = 0.0536) (Fig 6B). The increases in activated Arf6 levels in *Iqsec2* KO hemizygous and heterozygous mice were not reflected by a significant increase in Arf6 protein abundance relative to wild-type for either sex (detected in the same [and additional] cortical tissue) (Fig 6C). Nor was Arf6 protein abundance impacted by the presence of an observed seizure (indicated by stars instead of circles; *P* = 0.1009).

### Heterozygous loss-of-function variant in *IQSEC2* in a female with a neurocognitive seizure phenotype

Here, we report an elderly, 68-yr-old female (II-2) (Fig 7A) with severe-to-profound ID, early onset of seizures who is nonverbal,

and communicates primarily using sounds and gestures. She has a history of normal early development, learning to sit at 8–10 mo and walk at age 18 mo. After seizure onset at 17 mo, she developed repeated, generalised seizures during the daytime and regression was observed. She later developed additional atypical absence and atonic seizure types. Since the age of 11, she has been living in residential care. Her daily functioning skills are poor, and she needs assistance in every-day life. Facial features include low-set, large ears, and asymmetric facial features with prominent angle of the jaw, thick upper lip with mild hypertrichosis over the upper lip and deep-set eyes (Fig 7B). No brain MRI has been performed because of the requirement of general anaesthesia. A detailed clinical description of the proband is described in Supplemental Data 1. Whole-exome analysis of four Finnish families identified a novel, heterozygous single-nucleotide polymorphism at genomic position ChrX:g.53,349,756 (GRCh37/19) in the female proband of one family. The variant has been submitted to the gene variant database at https://databases.lovd.nl/shared/genes/IQSEC2 (patient ID 00174867; DB-ID #0000398628). Sanger sequencing confirmed the presence of this variant in the proband (Fig 7C). Samples were not available for the twin brother or parents of the affected female proband to confirm the inheritance status of this variant. This variant in exon 1 of the NM_001111125.2 long isoform substitutes a single nucleotide at c.566C>A, generating a predicted premature stop codon, p.(S189*)

(NP_001104595) in IQSEC2 (Fig 7D). This variant was not found in ExAC, GnomAD, or dbSNP150 project databases. This predicted premature stop codon is located 141 nucleotides from the exon 1–2 junction, with the transcript predicted to be degraded via the nonsense-mediated mRNA decay pathway, resulting in loss of the IQSEC2 protein.

Review of the literature shows that the affected female proband adds to the growing number of affected females with loss-of-function variants in IQSEC2 gene presenting with severe ID and early-onset seizures (Table 1) (29–42) recently reviewed (12). Table 1 details 31 different variants in IQSEC2 in 38 separate cases of affected female(s) predicted to cause pathogenic loss of IQSEC2 function. Of these 31 variants, 28 are known to have arisen de novo, one case of gonadal mosaicism in a family of four affected girls, and one case of monozygotic twins, with discordant phenotypes. The inheritance in the proband of the current report was unable to be determined. Although not all cases were accompanied by complete clinical descriptions, 33 of the 38 cases report developmental delay or ID ranging from mild-to-severe or profound and present with a range of comorbid behavioural and psychiatric features (Table 1). Interestingly, 28 of the 38 cases reported a range of seizure types in the affected females, including epileptic encephalopathies (Table 1). In a recent review of the phenotypic spectrum of epileptic encephalopathies in male and female patients with pathogenic IQSEC2 variants, it was noted that there was no specific electroclinical syndrome that could

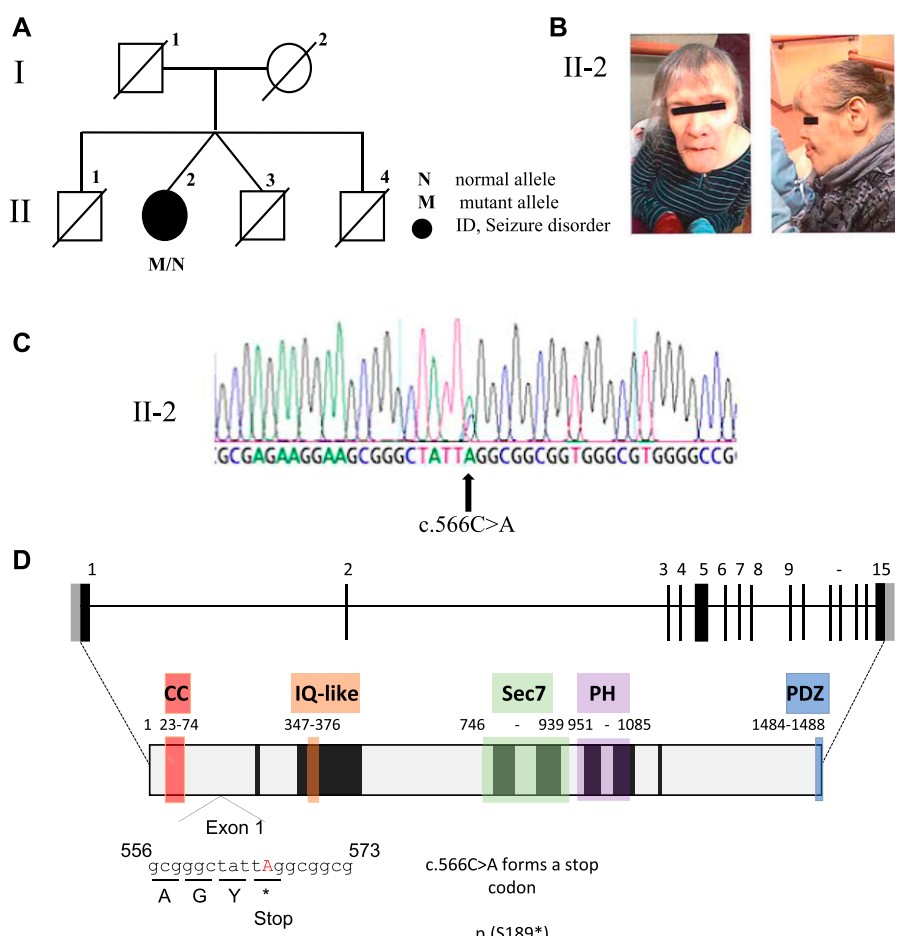

Figure 7.   Identification of a c.556C > A (NM_001111125.2) variant resulting in a premature stop codon at p.(S189*) (NP_001104595) in IQSEC2. (A) Pedigree of family. Open symbols represent unaffected individuals and filled black circle represents female with profound-to-severe intellectual disability and epilepsy. Normal (N) and mutant (M) alleles shown for proband. (B) Asymmetrical facial features, prominent angle of the jaw and low-set, large ears of II-2 (front and side). (C) DNA sequence electropherograms for the chrX: g.53349756 (GRCh37/hg19 assembly); c.556C>A mutation in exon 1 of 15 of IQSEC2 in II-2 affected female. (D) Predicted impact of novel variant in IQSEC2. The exon–intron structure of the longest isoform of the IQSEC2 gene (NM_001111125.2) with 15 exons, the ATG and open reading frame and stop codon position in black and 5′ and -3′ untranslated regions in light grey. The predicted protein structures (NP_001104595) with known functional domains highlighted; coiled–coiled (CC-red), IQ-like (orange) Sec7 enzyme domain (Green), PH domain (purple), and PDZ-binding motif (Blue), corresponding amino acids listed below each domain. The variant c.566C>A replaces the codon for Serine (p.189) for a stop codon and is predicted to result in nonsense-mediated mRNA decay and loss of the protein from the mutant allele.

**Table 1. Pathogenic loss-of-function variants in *IQSEC2* in females with intellectual disability and other comorbidities.**

| cDNA | Ex | Protein | Dom | Family | DD/ID | Seizures | Behavioural/Psychiatric/ Physical features | Ref |
|---|---|---|---|---|---|---|---|---|
| c.55_151delinsAT | 1 | p.Ala19Ilefs*32 | — | P1 | Mild ID | None | Speech deficits—pronunciation, syntax issues at 6.5 years. Tantrums, anxiety | (29) |
| c.83_85del | 1 | p.Asp28del | CC | 108286 | Rett like | None | Loss of language. Regression stabilization, gait abnormalities | (30) |
| c.273_282del | 1 | p.Asp91Lysfs*112 | — | P7 | Rett like | | Regression stabilization, gait abnormalities, stereotypic hand movements, inappropriate laughing/ screaming spells. Partial or loss of spoken language. | (31) |
| **c.566C>A** | **1** | **p.(S189*)** | **—** | **Fin2** | **Severe-profound ID** | **Generalised seizures (18 mo)** | **Limited speech, low-set large ears, asymmetric facial features, mild hypertrichosis, mild ASD.** | **This report** |
| c.804delC | 3 | p.Tyr269Thrfs*3 | — | 48 | | Seizures | Limited phenotype reported. | (32) |
| c.854del | 3 | p.Pro285Leufs*21 | — | P11 | Severe ID | Seizures (12 mo) tonic–clonic | Says words at 16 years. Limb rigidity, walking instability. | (29) |
| c.928G>T | 3 | p.Glu310* | — | P16 | Mild-mod DD | FE | No ASD or other features. Nonverbal at 3 years. | (33) |
| c.1556_1599delACCT | 5 | p.Tyr519Trpfs*87 | — | P10 | DD, Severe to profound ID | None | Hypotonia, first word at 2 years, stereotypies, and dysmorphic features. | (34) |
| c.1591C>T | 5 | p.Arg531* | — | P3 | DD, Severe to profound ID | Tonic–clonic, absence | Autism, first words at 11 mo, hypotonia, stereotypies, ataxic gait | (34) |
| c.1744_1763del | 5 | p.Arg582Cysfs*9 | — | P16 | Mild DD | Focal epilepsy (17 mo) | 50–60 words at 3 years. Autistic behaviour, hypertonia. | (29) |
| c.1983_1999del | 5 | p.Leu662Glnfs*25 | — | P17 | Global DD | Focal epilepsy (11 mo) | Babbling at 16 mo Hypertonia. | (29) |
| c.2052_2053delCG | 5 | p.Cys684* | — | 47 | | Seizures | Limited phenotype reported. | (32) |
| c.2078delG | 5 | p.Gly693Valfs*29 | | P18 | Mod global DD | None | Nonverbal at 2.8 years. Self-injurious behaviour, hypotonia | (29) |
| c.2203C>T | 5 | p.Gln735* | — | T17563 | Mild ID | SGE (5 years)—regression with nonconvulsive SE. Absence to tonic–clonic and myoclonic seizures, drop attacks. Offset at 38 years. | (35) | |
| c.2272C>T | 5 | p.Arg758* | — | P19 | Severe ID | Multifocal epilepsy (23 mo) | 3 words at 11.3 years. Self-injurious behaviours. | (29) |
| | | | | P20 | Mod ID | Seizures (9 years 4 mo) GTCS, focal, atypical absences | Speaks sentences, reasoning difficulties | |
| c.2317C>T | 6 | p.Gln773* | Sec7 | P6 | Global DD | Seizures (18 mo) | Hypotonic, strabismus, dysmorphic face | (36) |
| | | | | P23 | Mod ID | Seizures (14 years), GTCS, absences | Few words at 43 years. ASD (13 years) aggressive. | (29) |
| c.2317_2332del | 6 | p.Gln773Glyfs*25 | Sec7 | P24 | | Seizures (6 years) | Sentences at 11.3 years | (29) |

| cDNA | Ex | Protein | Dom | Family | DD/ID | Seizures | Behavioural/Psychiatric/ Physical features | Ref |
|------|----|---------|-----|--------|-------|----------|--------------------------------------------|-----|
| c.2679_2680insA | 8 | p.Asp894fs*10 | Sec7 | K2 | DD, Mod-severe ID | Epilepsy | 4 affected sisters, nonverbal (2), language delay (1), aggressive when young and ASD traits (2) | (37) |
| c.2776C>T | 9 | p.Arg926* | Sec7 | P3 | Severe ID Rett like | EE | ASD (balance & hand stereotypies), pain sensitivity & aggressive. Speech delay, regression at 2 years. Now nonverbal | (38) |
| | | | | P26 | Profound ID | LGS (23 mo) | Nonverbal at 11.3 years. Autistic behaviour, truncal hypotonia, strabismus. | (29) |
| | 9 | p.Tyr933* | Sec7 | M2189 | Global DD Mod ID | | ASD, sleep disturbances, behavioural aspects, oral motor dyspraxia, strabismus. Marked speech delay, nonverbal at 14 years. | (39) |
| c.2854C>T | 9 | p.Gln952* | PH | P27 | Severe ID | EE (12 years), absences, GTCS | Nonverbal at 16 years. Autistic behaviour, dystonia, tremor, ataxia. | (29) |
| c.2911C>T | 10 | p.Arg971* | PH | P8 | DD, Severe to profound ID | Seizures | No ASD. Stereotypies and dysmorphic features. | (34) |
| | | | | P11 | DD, Severe to profound ID | Seizures | Autism, first word at 2.3–3 years, stereotypies and dysmorphic features, ataxic gait | (34) |
| c.3079delC | 11 | p.Leu1027Serfs*75 | PH | P29 | Mod-severe ID | None | 10 words at 8 years | (29) |
| c.3163C>T | 12 | p.Arg1055* | PH | Pat19 | Severe ID | Epilepsy | Borderline macrocephaly, skewed X-inactivation (97:3) | (40) |
| | | | | P31 | Mod-severe ID | Seizures (5 years 8 mo) GTCS, focal dyscognitive | 3 word sentences, and 20 words at 8 years. Autistic behaviour, Global hypotonia, aggression, hyperactivity. | (29) |
| c.3278C>A | 13 | p.Ser1093* | — | P36 | Severe ID | None | Few words, rare sentences at 13 years. | (29) |
| c.3322C>T | 13 | p.Gln1108* | — | KO | | EE | | (41) |
| c.3433C>T | 13 | p.Arg1145* | — | P39 | Severe ID | Focal epilepsy (11 mo) focal, tonic, tonic–clonic | Nonverbal at 11 years. Autistic behaviour. | (29) |
| c.3457del | 14 | p.Arg1153Glyfs*244 | — | P40 | Severe ID | IS (7 mo) spasms, focal, absence, tonic, myoclonic jerks | Nonverbal at 20 years. Autistic behaviour, truncal hypotonia. MRI mild atrophy and cerebral white matter hyperintensities. | (29) |
| c.4039dupG | 15 | p.Ala1347Glyfs*40 | — | 1098 M | | EE (19 mo) | ASD, macrocephaly | (42) |
| | | | | P41 | Mild-mod ID | Seizures (3 years) absence and falls | Speaks sentences, writes first name, counts to 15 at 11 years. Mild autistic behaviour. | (29) |
| c.4401del | 15 | p.Gly1468Alafs*27 | — | P42 | Mod-severe ID | None | Short sentences at 11 years. Attention deficit/ hyperactivity | (29) |

**Table 1.** Continued

| cDNA | Ex | Protein | Dom | Family | DD/ID | Seizures | Behavioural/Psychiatric/Physical features | Ref |
|---|---|---|---|---|---|---|---|---|
| c.4419_4420insC | 15 | p.Ser1474Glnfs* | — | P6 | DD, Severe to profound ID | Absence, complex | Autism, hypotonia. First words at 7 years. Ataxic gait, stereotypies, bouts of laughter, self-injurious behaviour. | (34) |
| | | Twin sister of P6 | | P7 | DD, Mild ID | No | Autism, first words 11.5 mo. Ataxic gait | |

ASD, autistic spectrum disorder; DD, developmental delay; EE, epileptic encephalopathy; GTCS, generalised tonic–clonic seizures; IS, infantile spasms; SGE, symptomatic generalised epilepsy.

Del, deletion; dup, duplication. Numbers (alone) in brackets indicate number of affected individuals.

Nucleotide numbering reflects cDNA numbering with +1 corresponding to the A of the ATG translation initiation codon in the reference sequence for *IQSEC2* (GenBank: NM_001111125.2).

be defined, with all patients displaying multiple seizure types consisting mainly of atonic, myoclonic, or epileptic spasms. The seizure phenotypes were accompanied with a variety of electroencephalogram (EEG) patterning, including hypsarrhythmia, polyspikes and waves, generalised spikes and waves, slow spikes and waves, as well as background slowing (33). The advanced age of the female proband (68 yr old) we identify and present as part of this study underscores the importance of considering *IQSEC2* as an explanation of ID and, particularly, seizures in females across their life span. Furthermore, this finding reinforces the fact that a female *IQSEC2* phenotype may still be under-ascertained.

# Discussion

The cellular and molecular pathogenesis of *IQSEC2* mutations is not well understood. To address this, we used CRISPR/Cas9–targeted editing to generate an *Iqsec2* KO mouse model with no (*Iqsec2* KO hemizygous males) or reduced (*Iqsec2* KO heterozygous females) *Iqsec2* mRNA and protein. We validated both successful genome editing and germline transmission. To date, there have been no reports of transmission of loss-of-function mutations in *IQSEC2* in the human setting, with the exception of gonadal mosaicism in a family of four affected female siblings (37). We demonstrate that hemizygous KO males were viable and able to breed and generate healthy female heterozygous KO offspring. This suggests a potential difference in mouse to human phenotypic outcomes related to brain and sexual development (43, 44). Despite some difficulties, the successful breeding of the *Iqsec2* KO heterozygous female mice shows that a loss-of-function mutation in *Iqsec2* can be transmitted, at least in mice.

In agreement with the expanding phenotypic spectrum and the 67% penetrance of seizures in patients with de novo pathogenic variants in *IQSEC2* (33), we demonstrate that severe spontaneous seizures are observed in approximately half of all the *Iqsec2* KO mice. In these mice, we observed four distinct seizure types: (i) involuntary, uncontrolled, unilateral head movements, (ii) repetitive forelimb clonus, (iii) uncontrolled convulsions with bilateral forelimb outward stretching, and (iv) full body tonic–clonic seizures. Using the (human) International League Against Epilepsy classification system, these seizures can be considered equivalent to a (i) generalised clonic seizure, (ii) focal seizure evolving to

bilateral tonic–clonic seizure, (iii) focal motor seizure (likely frontal lobe onset), and (iv) generalised tonic–clonic seizure (Personal communication: A/Prof Nigel C Jones, Department of Neuroscience, Central Clinical School, Monash University, Melbourne, Victoria, Australia). Although EEG analysis was not undertaken in this study, aberrant firing and abnormal burst activity was noted in neuronal cultures of *Iqsec2* KO affected embryos. This abnormal activity is consistent with immature synaptic networks, previously associated with neurodevelopmental disorders (45, 46, 47).

Ionotropic glutamate receptors are ligand-gated cation channels that mediate most of the excitatory neurotransmission in the brain and are classified based on pharmacological selectivity to AMPA, kainic acid, and N-methyl-D-aspartic acid (NMDA). Activation of these receptors couples the electrical signal at the synapse to downstream biochemical signalling pathways. Selected NMDAR subunit genes have been implicated in the pathogenesis of ID (48, 49, 50). Synaptic plasticity associated with changes to spine morphology has been shown to be dependent on activation of these receptors (51, 52, 53, 54). IQSEC2 is localized to excitatory synapses as part of the protein scaffold downstream of the NMDA receptor complex. Through enzyme activity, IQSEC2 activates ARF6, a member of the Ras superfamily (55) to facilitate downstream remodelling of actin cytoskeleton, a site of convergence with other ID and autism genes. In the present study, we demonstrate that neurons in culture isolated from individual heterozygous female embryos display hallmarks of immature synaptic networks. Investigating which critical components of synaptic signalling are altered in response to Iqsec2 dosage, we show surprisingly the loss of Iqsec2 leads to an increase in the levels of activated Arf6 in cortical tissue during postnatal life. The increase in the active form of Arf6 is independent of changes to the overall abundance of Arf6 protein. Taken together, these data indicate that increased levels of activated Arf6 in the presence of Iqsec2 loss of function are likely to be at the local, dendritic environment in response to neuron signalling. Using the novel and powerful *Iqsec2* KO hemizygous male and heterozygous female mouse models will enable further investigations into the critical components altered in response to reduced Iqsec2 dosage and the downstream events contributing to disease outcomes.

Review of the female patients with complete loss-of-function mutations in *IQSEC2* shows that comorbid behavioural and

**Table 2. Correlation of behavioural findings in *Iqsec2* KO mice and patients with loss-of-function mutations.**

| Test | Measure | Finding | Patient trait |
|---|---|---|---|
| Inverted Grid | Neuromuscular strength | Reduced | Dystonia/stereotypic hand movements |
| Open Field | Exploration | Increased | Hyperactivity and psychiatric issues |
| | Anxiety (open spaces) | Increased | |
| Elevated zero maze | Fear response (height) | Reduced | Psychiatric issues |
| Y-maze | Short term memory | Reduced | Intellectual disability |
| Sociability | Social traits | Reduced | Autistic-like features |
| Barnes Maze | Cognition | Reduced | Intellectual disability |

psychiatric features are frequently present in addition to ID (Table 1). Hence, it was not surprising that mice modelling *Iqsec2* KO in the heterozygous female state displayed a range of phenotypic traits across a series of behavioural tests corresponding to these additional features (Table 2). Notably, the *Iqsec2* KO heterozygous females recapitulated a reduction in intellectual functioning and autistic-like behaviours, demonstrated through a loss of novel recognition on the Y-maze, a reduction in learning and memory during the Barnes maze trial, and an overall reduction in interaction time during the sociability test. Altered anxiety-like/fear responses and hyperactivity in the *Iqsec2* KO heterozygous female mice on multiple apparatus correlate with an increase in hippocampal volume, which has been associated with mental retardation and psychiatric issues such as autism, attention-deficit disorder, and schizophrenia (56, 57). We also observe a thinning of the corpus callosum in the *Iqsec2* KO heterozygous female mice, a phenotype emerging in several cases (29). Taken together, these findings highlight that our *Iqsec2* KO mouse model recapitulates the complex phenotypic spectrum observed in female patients with loss-of function *IQSEC2* variants (31, 32, 33, 35, 36, 37, 38, 39, 40, 41). Interestingly, the emergence of a speech phenotype is noted in the proband reported in this study and 26 of the 38 published female cases with loss-of-function variants (Table 1). Although we have not yet addressed this clinical feature in mice, it would be interesting to investigate it, particularly in view of the observations of reduced mothering skills of the breeding heterozygous females.

Female patients with de novo loss-of-function mutations in *IQSEC2* often have a more severe phenotype than the heterozygous state would traditionally predict, particularly if *IQSEC2* is thought to escape X-inactivation. The capacity of genes on the X-chromosome to be silenced, or to escape to some degree X-inactivation is not fully understood. Escape from X-inactivation for I*QSEC2* in humans has long been the prevailing view, as demonstrated by evidence measuring DNA methylation as a predictor of inactivation status across a panel of 27 tissues from 1,875 females (58). In contrast, recent large-scale expression studies from the GTEX consortium demonstrate that the expression of *IQSEC2* is similar in males and females across a broad range of tissues, including regions of the brain (19). This would suggest either dosage compensation in males (up-regulation) or females (down-regulation) or X-chromsome inactivation (XCI) in females. The degree by which incomplete XCI manifests as detectable sex differences in gene expression and phenotypic traits remains poorly understood (58, 59). The I*qsec2* KO

mouse model studied here shows severe phenotypic presentation in the heterozygous state. This fact would suggest that the severity of the phenotype in heterozygous females with loss-of-function IQSEC2/Iqsec2 allele is generally independent of X-chromosome inactivation. Given that X-inactivation leads to cellular mosaicism in heterozygous females one may speculate that the function and impact of IQSEC2/Iqsec2 is cell nonautonomous, that is, loss of IQSEC2/Iqsec2 impacts wild-type (i.e., where the mutant X is inactivated) as well as mutant (i.e., cells where the wild-type X is inactivated) cells. As such, this genetic KO model will provide an excellent tool to investigate the molecular mechanisms of X-linked inheritance underpinning the male and female phenotypes, providing valuable information not only for *Iqsec2*, but other X-linked genes with an emerging female phenotype (4, 5, 6, 37).

# Supplementary Information

# Acknowledgements

We would like to thank the members of the South Australian Genome Editing facility, Melissa White and Sandy Piltz, University of Adelaide, under the guidance of Professor Paul Q Thomas, for the expertise generating the *Iqsec2* KO mouse model. We would also like to thank the Laboratory Animal Services, Medical School South (Adelaide), and Aneta Zysk and Laura Redpath (Intellectual Disability Research Group) for their kind assistance with the mice, and Susan Hinze for her initial assistance with the CRISPR guide design, and Carl Campugan and Joel Chan for assistance in histology analysis and breakpoint mapping, respectively. Use of a laptop and behavioural tracking software, Anymaze (Wood Dale, United States of America), were kindly donated by Professor Bernhard Baune and Dr Catharine Jawahar, University of Adelaide. This research undertaken by the Intellectual Disability Research program in the Adelaide Medical School, University of Adelaide, Australia, was funded by the Australian National Health and Medical Research Council (Grant No 1063025) and Channel 7 Children's Research Foundation (Grant 161263). C Shoubridge was supported by the Australian Research Council (Future Fellowship FT120100086).

## Author Contributions

MR Jackson: investigation, formal anlaysis and writing—original draft.
KE Loring: investigation and writing—review and editing.

CC Homan: investigation, formal analysis.

MHN Thai: investigation.

L Määttänen: investigation.

M Arvio: investigation.

I Jarvela: investigation and writing—review and editing.

M Shaw: data curation.

A Gardner: investigation.

J Gecz: investigation, review and editing.

C Shoubridge: conceptualization, formal analysis, funding acquisition, investigation, and writing—original draft, review, and editing.

## Conflict of Interest Statement

The authors declare that they have no conflict of interest.

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
