## [Reviewer comments · Life Science Alliance]

Life Science Alliance

Heterozygous IQSEC2 loss increased activated Arf6 in severe female neurocognitive seizure phenotype

Matilda Jackson, Karagh Loring, Claire Homan, Monica Thai, Laura Määttänen, Maria Arvio, Irma Jarvela, Marie Shaw, Alison Gardner, Jozef Gecz, and Cheryl Shoubridge

DOI: <https://doi.org/10.26508/lsa.201900386>

Corresponding author(s): Cheryl Shoubridge, University of Adelaide, Adelaide Medical School

Review Timeline:

Submission Date:	2019-03-20
Editorial Decision:	2019-04-30
Revision Received:	2019-07-01
Editorial Decision:	2019-07-18
Revision Received:	2019-07-25
Accepted:	2019-08-15

Scientific Editor: Andrea Leibfried

Transaction Report:

April 30, 2019

Re: Life Science Alliance manuscript #LSA-2019-00386-T

Cheryl Shoubridge
University of Adelaide, Adelaide Medical School
Corner of George St and North Tce,
Adelaide
Australia

Dear Dr. Shoubridge,

Thank you for submitting your manuscript entitled "Heterozygous loss of function of IQSEC2 /Iqsec2 leads to increased activated Arf6 and severe neurocognitive seizure phenotype in females" to Life Science Alliance. The manuscript was assessed by expert reviewers, whose comments are appended to this letter.

As you will see, the reviewers appreciate your analysis of the Iqsec2 knock-out mice. They think, however, that the robustness of the results and the value of the mouse model need to get better demonstrated. We would thus like to invite you to submit a revised version, addressing the concerns of the reviewers. Importantly, we would expect a more careful discussion of the value of the mouse model and mentioning of the need to confirm the variant observed in a single patient in other patients. Furthermore, more brain anatomical data should get provided as well as histological data. Off-target effects should get ruled out to a commonly used level by sequencing potential off-targets (predicted off-targets based on the guide RNA used), while analysis of a second founder line is not mandatory for acceptance here. Potential compensatory roles of other BRAG family members should get analyzed and discussed as well. Reviewer #1 finds the male-specific mouse data underpowered and suggests to remove those. We think these data can get moved into the supplementary files and should get cautiously interpreted.

Thank you for this interesting contribution to Life Science Alliance. We are looking forward to receiving your revised manuscript.

Sincerely,

B. MANUSCRIPT ORGANIZATION AND FORMATTING:

Reviewer #1 (Comments to the Authors (Required)):

Jackson and colleagues have developed and characterised *Iqsec2* knockout mice using targeting CRISPR/Cas9 removal of exon 3. It provides an important model recapitulating most of the clinical manifestations observed in human patients and offers a powerful genetic tool to investigate the mechanisms of X-linked inheritance and X-linked genes with previously unrecognized phenotypes in females. The paper is very well-written. However, I have some concerns/questions.

1- Male versus female phenotypic characterization: the authors explain why it was challenging to obtain KO hemizygous male mice and therefore were unable to perform any behavioural testing in male mice. The authors however report some preliminary findings in KO hemizygous male mice for example in brain anatomy, MEA and levels of activated Arf6. Most are underpowered due to the small sample size (for example n=4 for MEA testing in males versus n=12 in females) which makes it hard to figure out whether there is or not differences between male and female mice. I am therefore not sure what the male data bring to the study. It might be more suited for a follow up study at a later stage when the breeding issues are resolved for example either using a conditional *Iqsec2* KO or changing the genetic background.

2- Inclusion of the loss of function variant in *IQSEC2* in a female patient: currently this section of the results almost reads as an anecdotal evidence. Can the mouse findings help in defining an assay that could be tested in the patient? For example, can Arf6 activation be tested in patient-derived cells to support the mouse findings? Otherwise, I am not sure what this case report brings to the result section of the paper. It could be moved to the discussion.

3- Brain anatomy: Considering the microcephaly and thin corpus callosum phenotypes described in patients (Mignot et al 2018), I wasn't sure why the focus was on the hippocampus (Figure 3h-i). Can the authors add a measurement of the total brain area and corpus callosum area? Arf6 activation is reported in the cortex. Is there an impact on the size of the cortex? Is there any reasons why Arf6 activation was not measured in the hippocampus? can the increased size of the dentate gyrus be linked back to increase Arf6 activation?

4- Generation of the model: Is the mouse reported here (I understand derived from founder A) identical (or not) to the previous paper (Hinze et al 2017) where the KO mice are used for primary neuronal cultures and assessment of cellular morphology? Also, to overcome potential off-target or compensatory effect between *IQSEC2* and *IQSEC3*, why was founder B not used instead as it shows the least effect on the expression of *IQSEC3* (14%)?

Reviewer #3 (Comments to the Authors (Required)):

Shoubridge's group first established the *IQSEC2* gene as a causative gene for intellectual disability with seizure, autistic traits and psychiatric problems (2010). In this study, they reported for the first time the phenotypes of mice lacking *Iqsec2* using the CRISPR/Cas9 system. The findings are potentially important to understand the mechanisms for clinical features observed in patients with *IQSEC2* syndrome. I have several concerns to be addressed before the manuscript is suitable for this journal.

(Major points)

- (1) Based on the data in Figure 1d, the *Iqsec3* gene seems to be directly or indirectly affected in founder mice. The reviewer is worrying about the off-target effects. The authors should confirm the phenotypes in an independent line. In addition, the *Iqsec3* gene should be sequenced to be intact.
 - (2) In Figure 1d, the protein levels of *Iqsec3* of each genotype should be quantified by immunoblotting. The protein levels of *Iqsec1* will also be informative to exclude the possibility of the compensatory roles of other BRAG family members.
 - (3) It is surprising that some phenotypes of heterogenous female mice are severe or more severe compared with those of hemizygous KO male mice. The reviewer wants to see the phenotypes of female homozygous mice. It would be informative to consider the sex-dependent *Iqsec2* functions.
 - (4) Previously, the authors' group reported that the manipulation of the *Iqsec2* expression level disturbed the neuronal morphology. Interestingly, the present histological assessment demonstrated the hippocampal volume was increased in female heterozygous mice. Histological assessments including the Nissl-staining of the whole brain and hippocampal region from three groups should be performed to show the readers whether mutant mice exhibit deficits in anatomical brain structures.
 - (5) In Figure 4h, are there any differences in volume of hippocampus between male and female control mice? Please provide information regarding the hippocampal volume (μm^3) of each genotype instead of the ratio (% age WT).
- (Minor points)
- (1) In Figure 1, DNA and protein marker sizes need to be included, especially when bands were cut.
 - (2) Similar to Figure 5c, the graphs in Figures 3 and 4 should present data (means {plus minus} SEM) of each genotype instead of the ratio (%age WT).
 - (3) In Figure 5, the label 'c' is lacking.
 - (4) In Figure 5, please provide information regarding the normalization of Arf6 protein.
 - (5) Typo: On page 4 line 84, 'IQSEC2 in' to 'IQSEC2 is'.
 - (6) On page 4 line 88, references #6 and #15 are inappropriately cited.

Re: Life Science Alliance manuscript #LSA-2019-00386-T

Dear Dr. Shoubridge,

Thank you for submitting your manuscript entitled "Heterozygous loss of function of IQSEC2 /Iqsec2 leads to increased activated Arf6 and severe neurocognitive seizure phenotype in females" to Life Science Alliance. The manuscript was assessed by expert reviewers, whose comments are appended to this letter.

As you will see, the reviewers appreciate your analysis of the Iqsec2 knock-out mice. They think, however, that the robustness of the results and the value of the mouse model need to get better demonstrated. We would thus like to invite you to submit a revised version, addressing the concerns of the reviewers. Importantly, we would expect a more **careful discussion of the value of the mouse model and mentioning of the need to confirm the variant observed in a single patient in other patients**. Furthermore, more **brain anatomical data** should get provided as well as **histological data**. **Off-target effects should get ruled out to a commonly used level by sequencing potential off-targets (predicted off-targets based on the guide RNA used)**, while analysis of a second founder line is not mandatory for acceptance here. Potential compensatory roles of other BRAG family members should get analyzed and discussed as well. Reviewer #1 finds the male-specific mouse data underpowered and suggests to remove those. We think these data can get moved into the supplementary files and should get cautiously interpreted.

To upload the revised version of your manuscript, please log in to your account: <https://lsa.msubmit.net/cgi-bin/main.plex>. You will be guided to complete the submission of your revised manuscript and to fill in all necessary information. Please get in touch in case you do not know or remember your login name. We would be happy to discuss the individual revision points further with you should this be helpful.

Thank you for this interesting contribution to Life Science Alliance. We are looking forward to receiving your revised manuscript.

- A letter addressing the reviewers' comments point by point.
- An editable version of the final text (.DOC or .DOCX) is needed for copyediting (no PDFs).
- High-resolution figure, supplementary figure and video files uploaded as individual files: See our

detailed guidelines for preparing your production-ready images, <http://www.life-science-alliance.org/authors>

B. MANUSCRIPT ORGANIZATION AND FORMATTING:

Reviewer #1 (Comments to the Authors (Required)):

Jackson and colleagues have developed and characterised Iqsec2 knockout mice using targeting CRISPR/Cas9 removal of exon 3. It provides an important model recapitulating most of the clinical manifestations observed in human patients and offers a powerful genetic tool to investigate the mechanisms of X-linked inheritance and X-linked genes with previously unrecognized phenotypes in females. The paper is very well-written. However, I have some concerns/questions.

1- Male versus female phenotypic characterization: the authors explain why it was challenging to obtain KO hemizygous male mice and therefore were unable to perform any behavioural testing in male mice. The authors however report some preliminary findings in KO hemizygous male mice for example in brain anatomy, MEA and levels of activated Arf6. Most are underpowered due to the small sample size (for example n=4 for MEA testing in males versus n=12 in females) which makes it hard to figure out whether there is or not differences between male and female mice. I am therefore not sure what the male data bring to the study. It might be more suited for a follow up study at a later stage when the breeding issues are resolved for example either using a conditional Iqsec2 KO or changing the genetic background.

Author Response: In response the reviewers request we have removed the MEA data from male WT and hemizygous KO mice from the report. However, we believe the data presented from the male mice is compelling and provides a useful comparison to highlight that the phenotype affects both sexes. As such we have retained the activated ARF6 data for both genders. As outlined in our responses below we have also provided additional histological and immunoblot analysis for both genders where adequately powered.

2- Inclusion of the loss of function variant in IQSEC2 in a female patient: currently this section of the results almost reads as an anecdotal evidence. Can the mouse findings help in defining an assay that could be tested in the patient? For example, can Arf6 activation be tested in patient-derived cells to support the mouse findings? Otherwise, I am not sure what this case report brings to the result section of the paper. It could be moved to the discussion.

Author Response: As we were reporting this novel variant in the elderly female patient for the first time, we thought to present the data as part of the results section. However, as suggested by this reviewer we have moved this section into the discussion, including reference to the figure and table.

3- Brain anatomy: Considering the microcephaly and thin corpus callosum phenotypes described in patients (Mignot et al 2018), I wasn't sure why the focus was on the hippocampus (Figure 3h-i). Can the authors add a measurement of the total brain area and corpus callosum area? Arf6 activation is reported in the cortex. Is there an impact on the size of the cortex? Is there any reasons why Arf6 activation was not measured in the hippocampus? can the increased size of the dentate gyrus be linked back to increase Arf6 activation?

Author Response:

We focused on measurements in the hippocampus as increases in hippocampal volume have been associated with mental retardation and psychiatric issues such as autism, attention-deficit disorder, and schizophrenia, the types of behavioural deficits detected as part of the studies in our *Iqsec2* KO Het female mice.

Our recent review of female patient phenotypes due to loss of function mutations (Shoubridge et al 2018) highlights there is limited reporting of MRI findings, likely due in part to difficulties in securing this type of data in severely intellectually disabled children / patients. Mignot and colleagues 2018, and a few other recent cases do report severely affected male patients having mild cerebral atrophy or thinning of the corpus callosum. However, this is not a consistent finding, and a review of patient data is warranted (but outside the scope of this report). In regard to the request for measures of brain and corpus callosum area / volume please refer to our responses to Reviewer 3, point 4 below.

We measured Arf6 activation in cortical tissue. *Iqsec2* is widely expressed across the cortical and cerebellar regions of the postnatal brain, not just the hippocampus. Using cortical tissue provided adequate biological sample to undertake the rigorous protein extraction method but ensuring enough volume resulted to test multiple replicates for each animal using the G-LISA assay format. We do not have any evidence to directly link the levels of activated ARF6 and increases in the size of the brain regions (please refer to the histological analysis added to the manuscript and the response below.

4- Generation of the model: Is the mouse reported here (I understand derived from founder A) identical (or not) to the previous paper (Hinze et al 2017) where the KO mice are used for primary neuronal cultures and assessment of cellular morphology? Also, to overcome potential off-target or compensatory effect between IQSEC2 and IQSEC3, why was founder B not used instead as it shows the least effect on the expression of IQSEC3 (14%)?

Author Response: The reviewer is correct that the mice were generated from Founder A which was (in a very condensed form) reported in Hinze et al 2017 (as mentioned in results). At the time of that report, we had not completed the extensive phenotyping of the mice, nor did we report the full details of CRISPR/Cas9 generation of the four founder lines. We believe this complete data is of interest to other researchers using this technology to generate gene-disease specific models. Hence, we have included this data as part of this report.

Progeny were not generated from Founder B as this mouse died at a very young postnatal time point, prior to reaching breeding age. In regard to potential compensatory effect of loss of *Iqsec2* by *Iqsec3*, although there is a slightly elevated level of expression in the founder mice, breeding of progeny across multiple generations clearly demonstrates that the levels of *Iqsec3* in these mice are not different to those in wild-type littermates, consistent with successful outbreeding of potential off-target or compensatory effects. Please see response to Reviewer 3 point 2 below for further analysis.

Reviewer #3 (Comments to the Authors (Required)):

Shoubridge's group first established the IQSEC2 gene as a causative gene for intellectual disability with seizure, autistic traits and psychiatric problems (2010). In this study, they reported for the first time the phenotypes of mice lacking *Iqsec2* using the CRISPR/Cas9 system. The findings are potentially important to understand the mechanisms for clinical features observed in patients with IQSEC2 syndrome. I have several concerns to be addressed before the manuscript is suitable for this journal.

(Major points)

(1) Based on the data in Figure 1d, the *Iqsec3* gene seems to be directly or indirectly affected in founder mice. The reviewer is worrying about the off-target effects. The authors should confirm the phenotypes in an independent line. In addition, the *Iqsec3* gene should be sequenced to be intact.

Author Response: We have addressed similar concerns in response to reviewer 1 – point 4. The levels of full length *Iqsec3* protein detected in progeny of the founder A mice clearly demonstrate similar expression to wild-type littermates indicating that the *Iqsec3* gene is intact.

(2) In Figure 1d, the protein levels of Iqsec3 of each genotype should be quantified by immunoblotting. The protein levels of Iqsec1 will also be informative to exclude the possibility of the compensatory roles of other BRAG family members.

Author Response: We agree the question of potential compensatory effect of loss of Iqsec2 by other family members is interesting. To address this, we quantified the levels of Iqsec3 and Iqsec1 protein by immunoblotting in all genotypes across postnatal life and demonstrate that there is no significant compensatory role by other members of the Iqsec2 family. We have updated the manuscript accordingly.

(3) It is surprising that some phenotypes of heterogenous female mice are severe or more severe compared with those of hemizygous KO male mice. The reviewer want to see the phenotypes of female homozygous mice. It would be informative to consider the sex-dependent Iqsec2 functions.

Author Response: The severity of the female phenotype does raise interesting questions regarding the sex-dependent function of Iqsec2. However, as we have not ever seen an affected female patient with a homozygous mutation in *IQSEC2*, we do not plan to pursue this line of inquiry in our mouse strain. As part of initial rounds of injections of our CRISPR/Cas9 strategy, we did generate one homozygous female founder that died early in postnatal life, preventing any robust follow-up or investigation.

(4) Previously, the authors' group reported that the manipulation of the Iqsec2 expression level disturbed the neuronal morphology. Interestingly, the present histological assessment demonstrated the hippocampal volume was increased in female heterozygous mice. Histological assessments including the Nissl-staining of the whole brain and hippocampal region from three groups should be performed to show the readers whether mutant mice exhibit deficits in anatomical brain structures.

Author Response: Nissl-staining of whole brain in both sagittal and coronal orientation for heterozygous female mice and wild-types have been added to the manuscript and supplementary data. In response to comments by Reviewer 1 we have also undertaken some measurement of brain area and the thickness of the corpus callosum in the female mice. We did not include data from the male mice considering the comments of Reviewer 1.

(5) In Figure 4h, are there any differences in volume of hippocampus between male and female control mice? Please provide information regarding the hippocampal volume (μm^3) of each genotype instead of the ratio (% age WT).

Author Response: Figure 3h has been updated as requested to show the hippocampal volume for each genotype instead of the ratio (%age WT) and illustrates there is no difference in volume between male and female control mice.

(Minor points)

(1) In Figure 1, DNA and protein marker sizes need to be included, especially when bands were cut.

Author Response: We have listed the amplicon size or predicted protein band size on each panel and show the DNA marker sizes.

(2) Similar to Figure 5c, the graphs in Figures 3 and 4 should present data (means {plus minus} SEM) of each genotype instead of the ratio (%age WT).

Author Response: The appropriate graphs in Figure 3 and 4 have been adjusted as requested.

(3) In Figure 5, the label 'c' is lacking.

Author Response: Thank you for picking this up. The label has been added.

(4) In Figure 5, please provide information regarding the normalization of Arf6 protein.

Author Response: The levels of activated Arf6 are measured in protein lysates extracted for and measured by a G-LISA assay (Figure 5b). The Arf6 protein abundance (Figure 5c) was measured in protein lysates requiring a different extraction method using immunoblot and subsequent analysis. As such, this data was not normalised against each other. The text relating to figure 5 has been clarified.

(5) Typo: On page 4 line 84, 'IQSEC2 in' to 'IQSEC2 is'.

Author Response: This has been corrected.

(6) On page 4 line 88, references #6 and #15 are inappropriately cited.

Author Response: Thank you for picking up this error in our endnote formatting. The correct references of Myers et al 2012 and Brown et al 2016 have replaced the incorrectly cited references.

July 18, 2019

RE: Life Science Alliance Manuscript #LSA-2019-00386-TR

Prof. Cheryl Shoubridge
University of Adelaide, Adelaide Medical School
Corner of George St and North Tce,
Adelaide 5000
Australia

Dear Dr. Shoubridge,

Thank you for submitting your revised manuscript entitled "Heterozygous IQSEC2 loss increased activated Arf6 in severe female neurocognitive seizure phenotype". As you will see in the reports copied below, a few issues remain that need to get addressed in a further revision. The manuscript files also need to adhere to our formatting guidelines to allow publication here, which will require a very careful revision. We would thus like to invite you to submit final files to us. Importantly:

- Please address the remaining comments of reviewer #1 and #3. For point 2 of rev#3, please move the patient data into the results section (currently in discussion)
- Please provide documentation about the institutional approval for the human data and that they conform with the principles set out in the WMA Declaration of Helsinki and the Department of Health and Human Services Belmont Report
- Please deposit and provide the accession code for the whole exome seq data
- The manuscript text needs to get provided in word docx format
- All figure panels (a, b, c etc) need to get individually mentioned in the manuscript text
- The "data file" texts should get included in the main manuscript to allow readers to fully appreciate the content of your manuscript
- The references can remain numerical, but should be in square brackets (not superscript as currently done)
- All figures (also S figures) need to get uploaded as separate files, and the figures should not span across multiple pages (as currently the case for Fig 4)
- Some panels miss labeling (eg. Panel 'b' in Fig 4, panel 'a' in Fig S2), please add
- A callout for Table S2 is missing from the manuscript text, please add
- Fig S6 misses panel c and d, please correct

A. FINAL FILES:

B. MANUSCRIPT ORGANIZATION AND FORMATTING:

Thank you for your attention to these final processing requirements.

Sincerely,

Andrea Leibfried, PhD
Executive Editor

Life Science Alliance
Meyerhofstr. 1
69117 Heidelberg, Germany
t +49 6221 8891 502
e a.leibfried@life-science-alliance.org
www.life-science-alliance.org

Reviewer #1 (Comments to the Authors (Required)):

The authors have been responsive to the previous round of comments and I appreciate the effort that has gone into their revised manuscript. I have a few remaining comments based on the revised manuscript. My main concern is with new Figure 4 and the neuroanatomical studies. To help the readers, can the neuroanatomical findings (including illustration images and quantifications) be grouped all together in this new Figure 4 (currently the hippocampus related measurements are presented in Figure 3, its associated-image in new Supplementary Figure 6 and the total brain area in new Supplementary Figure 6). I think it is fine to have this new Figure 4 dedicated to the coronal plane and new Supplementary Figure 6 to sagittal, but new Figure 4 is currently incomplete and should gather the hippocampus related measures as well as the total brain area in coronal plane. Please review the legend of new Supplementary Figure 6 (there are discrepancies between the number of panels in the legend compared to the figure). Also, I wasn't sure what the authors meant by "the thickness of the CC was measured where the start and the end of the cingulate intercept ..." did the authors meant the cingulum (not the cingulate)? I have a comment on measurement 3 of the CC which rather corresponds to the thickness of the external capsule. Can this be replaced by the measurement of the area of the corpus callosum. Finally, and most importantly, please mention the position of the brain sections (coronal and sagittal) relative to the reference Mouse Brain Atlas (Paxinos and Franklin) as the size of the corpus callosum and the size of the hippocampus are brain regions that significantly vary in size depending on the position (when anterior or posterior to the chosen section of interest). Groups of animals should therefore be analyzed at equivalent position.

Reviewer #3 (Comments to the Authors (Required)):

I thank the authors for their extensive revision in response to the reviewers' comments. My remaining concerns are as follows:

- (1) Since the seizure could have some impact on the brain morphology, the information on whether or not mice used for morphological analyses in Fig. 3h and 4 had spontaneous seizure before sacrifice should be included like Fig. 2e and 6.
- (2) The revised discussion is very peculiar for me, because the discussion should be basically done based on the results. Therefore, I gently suggest that this novel variant in an elderly female be excluded in this manuscript and reported elsewhere.

RE: Life Science Alliance Manuscript #LSA-2019-00386-TR

"Heterozygous IQSEC2 loss increased activated Arf6 in severe female neurocognitive seizure phenotype".

Matilda R Jackson^{1,2}, Karagh E Loring^{1,2}, Claire C Homan², Monica HN Thai¹, Laura Määttä³, Maria Arvio^{3,4,5}, Irma Jarvela⁶, Marie Shaw², Alison Gardner², and Jozef Gecz^{2,7}, Cheryl Shoubridge^{1,2*}

Summary of required actions as listed in email dated 18 July 2019.

- Please address the remaining comments of reviewer #1 and #3. For point 2 of rev#3, please move the patient data into the results section (currently in discussion)

Author response: Please find our point-by-point response to reviewers' comments below. We happily have moved back the patient data into the results. This section was only moved this into the discussion in response the first round of reviewers' comments.

- Please provide documentation about the institutional approval for the human data and that they are conform with the principles set out in the WMA Declaration of Helsinki and the Department of Health and Human Services Belmont Report

Author response: We have confirmed with our Human Ethics committee the following; Researchers in Australia are responsible for undertaking research that is designed and conducted in accordance with the *Australian Code for the Responsible Conduct of Research (updated 2018)* and the *National Statement on Ethical Conduct in Human Research (2018)*, in addition to other (Commonwealth, State and Local) jurisdictional obligations.

The Preamble of the *National Statement (2018)* takes into account the *Declaration of Helsinki*, hence we can say **Yes** to that question that we have "*conformed with the principles set out in the WMA Declaration of Helsinki*".

However, as an Australian jurisdiction, we do not conform to the USA's *Belmont Report (1979)*, but indirectly the principles in that Report are covered by the *National Statement (2018)*. Hence, it is our contention that our *National Statement (2018)* is equivalent for the Australian research context regarding ethical principles for human research.

The information regarding our institutional approval was provided in the materials and methods section. We have updated this section with our approval number. **This section now reads;**

"The screening protocols were approved by the Women's and Children's Health Network Human Research Ethics Committee and the Human Ethics Committee of The University of Adelaide, Adelaide, Australia (approval number REC2361/03/2020) and conforms with the principles set out in the WMA Declaration of Helsinki and Australian National Statement on Ethical Conduct in Human Research (2018). Informed consent was obtained from carers of the patient, including consent to publish images."

- Please deposit and provide the accession code for the whole exome seq data

Author response: We can provide this data upon request. It is not our practice to upload and make publicly available the sequencing data from clinical samples screened as part of our research activities.

- The manuscript text needs to get provided in word docx format

Author response: The manuscript is provided as a docx format, with all figures removed.

- All figure panels (a, b, c etc) need to get individually mentioned in the manuscript text

Author response: All figure panels have been individually mentioned in the manuscript.

- The "data file" texts should get included in the main manuscript to allow readers to fully appreciate the content of your manuscript

Author response: We have updated the manuscript with as much of the description of data from the "data file" without reducing the clarity and flow of the main document. There remains one Supplementary data file outlining a more extensive patient clinical description.

- The references can remain numerical, but should be in square brackets (not superscript as currently done)

Author response: Thank you for this guidance. We have kept the numerical format, but have edited to comply with journal guidelines.

- All figures (also S figures) need to get uploaded as separate files, and the figures should not span across multiple pages (as currently the case for Fig 4)

Author response: We have uploaded all figures as separate files. In regard to Figure 4, to ensure that the picture panels are not too small, we have removed some of the panels.

- Some panels miss labelling (eg. Panel 'b' in Fig 4, panel 'a' in Fig S2), please add

Author response: We have updated and checked all figures and ensure all panels are correct and labelled.

- A callout for Table S2 is missing from the manuscript text, please add

Author response: Table S2 is referenced in the materials and methods section headed "*Genotyping*". As this appears prior to the call out for Table S1, we have updated the Supp table numbering (in main text and in File name) and uploaded each of these Supplementary tables as xls files.

- Fig S6 misses panel c and d, please correct

Author response: Our apologies, the additional panels were added into a main manuscript figure instead of this supplementary figure. We have adjusted the figure legend in Fig S6 accordingly.

Editorial Requirements: as listed in the email have all been considered and where appropriate undertaken.

Reviewers comments:

Reviewer #1 (Comments to the Authors (Required)):

The authors have been responsive to the previous round of comments and I appreciate the effort that has gone into their revised manuscript. I have a few remaining comments based on the revised manuscript. My main concern is with new Figure 4 and the neuroanatomical studies. To help the readers, can the neuroanatomical findings (including illustration images and quantifications) be grouped all together in this new Figure 4 (currently the hippocampus related measurements are presented in Figure 3, its associated-image in new Supplementary Figure 6 and the total brain area in new Supplementary Figure 6). I think it is fine to have this new Figure 4 dedicated to the coronal plane and new Supplementary Figure 6 to sagittal, but new Figure 4 is currently incomplete and should gather the hippocampus related measures as well as the total brain area in coronal plane.

Author response: Thank you for the additional suggestions. We have consolidated the neuroanatomical findings into one figure in the manuscript. Hippocampal volume was measured in sagittal sections, whilst the thickness of corpus colosum was measured in coronal sections. We have combined neuroanatomical measures together in Figure 4.

Please review the legend of new Supplementary Figure 6 (there are discrepancies between the number of panels in the legend compared to the figure). Also, I wasn't sure what the authors meant by "the thickness of the CC was measured where the start and the end of the cingulate intercept ..." did the authors meant the cingulum (not the cingulate)? I have a comment on measurement 3 of the CC

which rather corresponds to the thickness of the external capsule. Can this be replaced by the measurement of the area of the corpus callosum.

Author response: Our apologies. We have combined neuroanatomical measures together in Figure 4. We have removed the third measurement of thickness of the CC, and corrected the terminology.

Finally, and most importantly, please mention the position of the brain sections (coronal and sagittal) relative to the reference Mouse Brain Atlas (Paxinos and Franklin) as the size of the corpus callosum and the size of the hippocampus are brain regions that significantly vary in size depending on the position (when anterior or posterior to the chosen section of interest). Groups of animals should therefore be analyzed at equivalent position.

Author response: We confirm that all comparisons between groups were conducted at equivalent positions and have supplied relative to the reference mouse brain atlas included in the results sections where appropriate.

Reviewer #3 (Comments to the Authors (Required)):

I thank the authors for their extensive revision in response to the reviewers' comments.

My remaining concerns are as follows:

(1) Since the seizure could have some impact on the brain morphology, the information on whether or not mice used for morphological analyses in Fig. 3h and 4 had spontaneous seizure before sacrifice should be included like Fig. 2e and 6.

Author response: We have included the seizure data into the graphs as requested. Given the sporadic nature of the seizures in these mice, even in Het KO female mice in which we have not observed a seizure we cannot rule out that a seizure(s) may have occurred when not being observed.

(2) The revised discussion is very peculiar for me, because the discussion should be basically done based on the results. Therefore, I gently suggest that this novel variant in an elderly female be excluded in this manuscript and reported elsewhere.

Author response: We have happily moved this data back into the results. This section was only moved this into the discussion in response the first round of reviewers' comments.

August 15, 2019

RE: Life Science Alliance Manuscript #LSA-2019-00386-TRR

Prof. Cheryl Shoubridge
University of Adelaide, Adelaide Medical School
Corner of George St and North Tce,
Adelaide 5000
Australia

Dear Dr. Shoubridge,

Thank you for submitting your Research Article entitled "Heterozygous IQSEC2 loss increased activated Arf6 in severe female neurocognitive seizure phenotype". It is a pleasure to let you know that your manuscript is now accepted for publication in Life Science Alliance. Congratulations on this interesting work.

DISTRIBUTION OF MATERIALS:

Again, congratulations on a very nice paper. I hope you found the review process to be constructive and are pleased with how the manuscript was handled editorially. We look forward to future exciting submissions from your lab.

Sincerely,
